# CoSeLECT: Adaptive Frame Selection for Video-Language Understanding

## Abstract

Multimodal Large Language Models (MLLMs) have shown strong performance on image understanding tasks, but video comprehension remains a significant challenge due to the high computational cost of processing long frame sequences and the limited token capacity of underlying Large Language Models (LLMs). Prior approaches to address this often rely on uniform frame sampling, query-agnostic pruning, or require costly training of dedicated compression modules. In this work, we introduce CoSeLECT, a training-free, plug-and-play, query-guided frame selection method that intelligently subsamples video frames for efficient use in MLLMs. CoSeLECT leverages two key signals: temporal redundancy, which identifies similar frame clusters, and query relevance, which selects frames based on their semantic alignment with the input query. By combining these signals through an adaptive frame selection strategy, CoSeLECT selects frames that are both diverse and highly relevant to the query, without requiring any model-specific tuning. Our results on various base MLLMs show that CoSeLECT consistently outperforms trained and training-free state-of-the-art methods, including LongVU by +3.8% on MLVU and AKS by +4.5% on EgoSchema.

## 1 Introduction

Multimodal Large Language Models (MLLMs) (Liu et al., 2023; Li et al., 2023b; Shi et al., 2025a) have shown strong performance on image comprehension and reasoning tasks (Antol et al., 2015; Vinyals et al., 2015). The standard paradigm projects images into embeddings aligned with a Large Language Model (LLM)'s token space, enabling the LLM to leverage its reasoning capabilities. While effective for images, scaling to videos is far more challenging: hundreds or thousands of frames yield prohibitively many visual tokens. For instance, LLaVA-OneVision (Li et al., 2024a), despite its 32K token context window (Team, 2024), can only accommodate ∼160 frames per pass, since each frame produces around 200 tokens. Thus, video understanding performance critically depends on which frames are selected.

A large body of work addresses this token bottleneck. The default remains naive *input-blind* strategies such as uniform or fixed fps sampling (Li et al., 2024a; Bai et al., 2025b), which often overload the LLM with redundancy while missing informative moments. More sophisticated compression techniques prune or merge tokens based on spatial or temporal redundancy (Shang et al., 2024; Chen et al., 2024a; Shen et al., 2025; He et al., 2024; Fu et al., 2024b), but these remain query-agnostic. Query-aware methods exist, but typically require costly training (Li et al., 2024d; Shen et al., 2024) or rely on intermediate attention maps (Shi et al., 2025b; Huang et al., 2024), limiting plug-and-play usability.

**Frame selection** has emerged as a more interpretable approach to token reduction, targeting redundancy at the level of entire frames. Existing training-free methods such as MIF and MDF (Han et al., 2024) provide plug-and-play compatibility, but leave much of the available signal unused: MIF operates indirectly in caption space, MDF is entirely query-agnostic, and overall these approaches lag behind trained methods. KeyVideoLLM (Liang et al., 2024) introduces query guidance, but does not explicitly model temporal which can be useful for video understanding. In contrast, trainable methods (Wang et al., 2024a; Yu et al., 2023; Wang et al., 2024b; Yu et al., 2025) learn to combine signals more effectively and achieve stronger results, but at the cost of added complexity and training overhead. *Crucially, these heavier methods are typically limited to sparsely pre-sampled frame pools*

*in order to remain computationally feasible—risking the permanent loss of "needle-in-a-haystack" moments before the selection algorithm can even evaluate them, a limitation that becomes particularly acute under resource constraints.*

From these trends, we identify three desiderata for an effective frame selection module in MLLMs: (1) adaptivity to both video content and query, (2) training-free, plug-and-play operation for easy deployment, and (3) scalability to large candidate pools without excessive overhead. *Meeting these desiderata motivates a return to first principles: rather than building increasingly complex modules, the challenge is to **effectively** fuse simple, efficiently computable signals in a way that scales.*

We propose CoSeLECT (**Co**ntinuity-aware **Se**mantic **L**ocalization and **E**xtraction of **C**andidate **T**okens), a training-free, query-guided frame selection algorithm designed around this principle. The novelty of CoSeLECT lies not in introducing new signals, but in its lightweight, principled fusion of two readily available ones—frame–text similarity for semantic relevance and inter-frame similarity for temporal continuity. By combining them in a single-stage allocation scheme with duration-aware weighting, CoSeLECT achieves stable, interpretable selections that scale to large candidate pools. This simplicity enables both plug-and-play use and improved performance as the pool grows: from narrow queries (e.g., "What color is the car that passes by home?") to broad temporal ones (e.g., "What activity was the child doing in the park?"), CoSeLECT consistently identifies compact yet comprehensive frame sets, and is shown to outperform even strong trained methods such as LongVU.

We evaluate CoSeLECT on six video understanding benchmarks—VideoMME (Fu et al., 2024a), MLVU (Zhou et al., 2024), MVBench (Li et al., 2024c), EgoSchema (Mangalam et al., 2023), LongVideoBench (Wu et al., 2024), and Next-QA (Xiao et al., 2021). Across all settings, CoSeLECT outperforms or matches state-of-the-art methods, both training-free and fine-tuned, **demonstrating the power of scaled simplicity**.

**Our contributions are summarized as follows:**

- We introduce CoSeLECT, a training-free, query-guided frame selection algorithm with a novel fusion strategy that explicitly balances semantic relevance and temporal continuity.

- We show that training-free CoSeLECT consistently outperforms or matches state-of-the-art methods across six video understanding benchmarks, including gains of +3.8% over LongVU (Shen et al., 2024) (trained) on MLVU and +4.5% over AKS (Tang et al., 2025a) (training-free SOTA) on EgoSchema.

- We demonstrate that CoSeLECT generalizes robustly across architectures (LLaVA-OneVision, Qwen2.5-VL) and model scales (0.5B/7B) without model-specific tuning, and provide comprehensive ablations validating the contribution of each design choice.

## 2    RELATED WORKS

### 2.1    FROM IMAGES TO VIDEOS: THE CHALLENGE OF TOKEN EFFICIENCY

Following the success of Multimodal Large Language Models (MLLMs) in image comprehension (Liu et al., 2023; Zhu et al., 2023; Li et al., 2023a; Dai et al., 2023), the field has naturally progressed towards adapting these architectures for video understanding. Initial works extended popular image-based frameworks by, for example, feeding pooled frame features into the LLM (Maaz et al., 2024; Xu et al., 2024a), incorporating audio signals via a trained Q-Former (Zhang et al., 2023), or pre-aligning image and video encoders (Lin et al., 2024).

However, a primary challenge in this transition is the immense computational cost associated with processing a large sequence of visual tokens. Models that uniformly sample dozens of frames (Li et al., 2024a) face a significant inference burden due to the quadratic complexity of self-attention. To mitigate this, various **token reduction** strategies have been proposed. These include methods that prune visual tokens based on low attention scores (Chen et al., 2024a; Huang et al., 2024), condense frame information into representative tokens (Li et al., 2024d), or employ specialized pooling pathways (Xu et al., 2024b). Among these strategies, a particularly effective and interpretable approach is **keyframe selection**, which focuses on identifying the most salient full frames rather than individual tokens.

## 2.2 Approaches to Keyframe Selection

**Training-free methods** offer desirable plug-and-play compatibility. Some, like MIF and MDF (Han et al., 2024), are not directly comparable to our setting: MIF operates indirectly in caption space by ranking generated descriptions, while MDF is entirely query-agnostic, prioritizing only visual diversity. KeyVideoLLM (Liang et al., 2024) adopts a two-stage coarse-to-fine pipeline but does not explicitly model temporal continuity, and its code has not been released, preventing direct comparison. Other recent training-free approaches include SlowFast-LLaVA (Xu et al., 2024b), which pools tokens through slow and fast pathways; Static-or-Dynamic (Shi et al., 2025b), which explores candidate subsequences and selects based on the query; and QuoTA (Luo et al., 2025), which augments the query with chain-of-thought reasoning before scoring frames. Adaptive Keyframe Sampling (AKS) (Tang et al., 2025a) remains the most comparable baseline, applying a recursive divide-and-conquer procedure over frame–text similarity scores. As AKS is the strongest existing training-free method, we contrast our algorithmic design in Section B.6 and show consistent gains for CoSeLECT in Table 1.

**Trainable methods** introduce learnable modules for frame selection, such as Gaussian masks under weak supervision (Wang et al., 2024a), cross-modal distillation with a "Frame-Prompter" (Wang et al., 2024b), lightweight heads trained with pseudo-labels (Yu et al., 2023), or ranking-based supervision of frame combinations (Yu et al., 2025). While effective, they add components and training overhead, limiting flexibility and plug-and-play use with off-the-shelf MLLMs.

By jointly modeling semantic relevance and temporal diversity in a single-stage adaptive procedure, CoSeLECT achieves efficiency, context-awareness, and broad compatibility without additional supervision.

## 3 Method

We introduce **CoSeLECT** (**Co**ntinuity-aware **Se**mantic **L**ocalization and **E**xtraction of **C**andidate **T**okens), a query-aware frame selection method for understanding long videos with multimodal large language models. A central challenge in video-based LLM reasoning lies in efficiently utilizing a limited token context window when processing temporally extended visual content. Our approach addresses this by selectively identifying frames that are both semantically relevant to the query and representative of the video's visual narrative. By balancing these two complementary signals—textual relevance and visual continuity—we aim to maximize the informational value of each selected frame while remaining within token constraints. Unlike uniform sampling, which implicitly assumes that temporal regularity correlates with information value, our method dynamically adapts frame selection to the content and query, reducing redundancy and improving downstream reasoning.

### 3.1 Setting the Stage

Multimodal Large Language Models (MLLMs) are subject to strict token limits, which makes aggressive frame sampling essential for processing video inputs. For example, a 3-minute video sampled at 1 frame per second yields 180 frames—yet even advanced models like LLaVA-OneVision (Li et al., 2024a) can accommodate only about 160 frames within a 32K token context window (Team, 2024), since each frame typically consumes 200 tokens.

The standard MLLM video pipeline typically proceeds as follows: (1) a video and query are provided, (2) frames are uniformly sampled (e.g., at 1 fps), (3) each frame is encoded via a visual encoder, and (4) the resulting embeddings are passed—alongside the query—into the LLM. Crucially, this process is query-agnostic at the sampling stage and assumes uniform temporal coverage is sufficient. In practice, this often leads to the selection of semantically uninformative frames and results in suboptimal use of the model's limited token budget.

We propose a query-aware alternative to this sampling step that adaptively selects frames based on both their semantic relevance to the query and their contribution to the video's overall visual continuity. Our approach comprises three key components: a *Text Relevance Signal* that estimates frame-query alignment, a *Visual Continuity Signal* that detects coherent temporal segments, and an *Adaptive Frame Selection* strategy that integrates these signals to effectively allocate the limited frame budget.

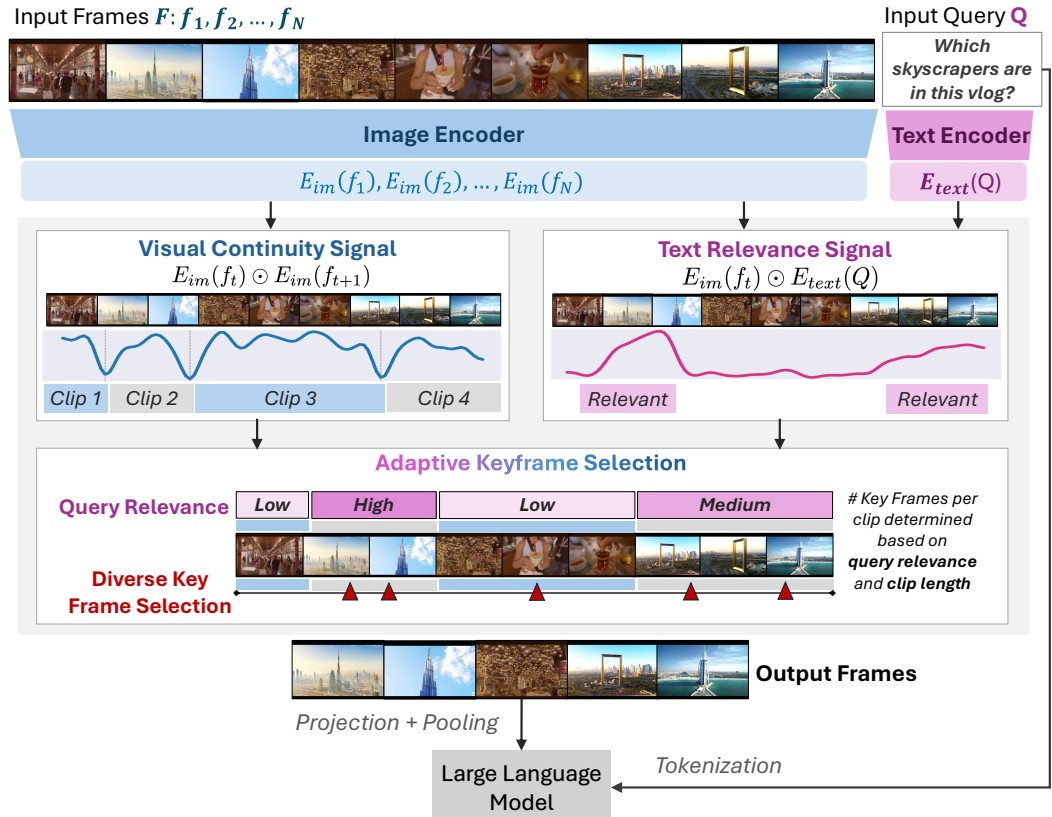

Figure 1: **CoSeLECT**: A query-aware frame selection method that identifies frames which are both **semantically relevant** to the query and **representative** of the video's visual narrative for understanding long videos with token budget constrained multimodal language models.

## 3.2 TEXT RELEVANCE SIGNAL

Some frames in a video are substantially more relevant to the user's query than others. A query-aware sampling method must therefore estimate semantic alignment between the query and each frame to ensure the most informative content is retained. This signal forms the core of our relevance-driven sampling strategy.

We begin by uniformly sampling the input video into $N$ candidate frames $f_1, f_2, ..., f_N$, from which we aim to select $K$ frames to send to the LLM, where $N > K$. For clarity, in the tables we denote $N$ as pre-$\mathbf{E}_{im}$ and $K$ as post-$\mathbf{E}_{im}$, corresponding to the number of image embeddings before and after selection, respectively. as Each frame is embedded using SigLIP-So400M-patch14-384 (Zhai et al., 2023), a vision-language model that produces aligned image-text representations. Since the finetuned SigLIP is also used as the vision encoder in LLaVA-OneVision (Li et al., 2024a), these embeddings can be directly reused, avoiding redundant computation. (Worth noting that this doesn't hold true if you swap out vision encoders.) The image and text encoders, $E_{im}(\cdot)$ and $E_{text}(\cdot)$ respectively, are used to compute a relevance score for each frame.

For a given query $Q$, we compute cosine similarity between each frame and query embeddings:

$$\text{S}_{\text{text}}(t) = E_{im}(f_t) \odot E_{text}(Q) \tag{1}$$

where $\odot$ denotes cosine similarity between normalized embedding vectors. High values of $\text{S}_{\text{text}}(t)$ indicate that the frame is semantically aligned with the query. To reduce noise, we apply Gaussian smoothing across the temporal dimension, ensuring more stable relevance scores and reducing sensitivity to abrupt local fluctuations.

## 3.3 Visual Continuity Signal

Focusing solely on query relevance results in redundancy or neglecting important temporally contextual segments. To encourage broad and meaningful coverage of the video, we introduce a second signal that identifies scene transitions and partitions the video into visually coherent segments.

This signal is based on the intuition that sudden visual changes often correspond to new scenes or shifts in content. We compute cosine similarity between consecutive frame embeddings:

$$S_{\text{frame}}(t) = E_{im}(f_t) \odot E_{im}(f_{t+1}) \tag{2}$$

After applying Gaussian smoothing to incorporate local context, we detect visual discontinuities by identifying significant drops in similarity:

$$\text{SceneBoundaries} = k \mid S_{\text{frame}}(k) < \tau_{\text{sim}} \tag{3}$$

with a threshold $\tau_{\text{sim}} = 0.8$, chosen via empirical validation 8. These boundaries segment the video into subclips $C = C_1, ..., C_M$, each representing a coherent visual scene. This helps ensure our frame selection covers the full narrative arc of the video rather than focusing narrowly on isolated high-relevance moments. As our ablations 4.3(see *Frame Selection Strategy*) show, this continuity prior plays a crucial role in maintaining context necessary for strong video understanding.

## 3.4 Adaptive KeyFrame Selection

While the text relevance and visual continuity signals offer powerful tools individually, leveraging them in isolation risks suboptimal results. Focusing only on relevance may oversample redundant moments (e.g., repeating frames in a key scene), while emphasizing visual coverage alone may dilute attention across low-information segments. The challenge, then, is to **adaptively allocate limited frame tokens to scenes that matter most—those that are both content-rich and semantically aligned with the query—while ensuring diverse temporal coverage**.

We achieve this by designing a selection strategy that fuses these two signals and distributes the frame budget in proportion to the estimated importance of each visual subclip.

For each identified subclip $C_i$, we compute a composite relevance score that captures both standout moments and overall alignment with the query:

$$R_i = \max(S_{\text{text}}|C_i) + \text{mean}(S_{\text{text}}|C_i) \tag{4}$$

The maximum highlights peak relevance within the subclip, while the mean ensures we value consistent alignment across its duration. This helps us select subclips that are either uniformly relevant or contain particularly important frames. We empirically verify the importance of this design choice in 7

We then weight this relevance score by the square root of the subclip's duration $D_i$:

$$W_i = R_i \cdot \sqrt{D_i} \tag{5}$$

This duration-adjusted weighting ensures longer relevant segments receive more frames than shorter ones, but the relationship is sublinear (square root). Without this adjustment, extremely long subclips would dominate the frame budget regardless of content relevance, while using the square root prevents this overrepresentation while still acknowledging that longer segments typically contain more diverse content in comparison to shorter ones. We empirically verify the importance of this design choice in 7

The frame budget of $K$ total frames is distributed across subclips proportionally to these weights:

$$k_i = \lfloor K \cdot \frac{W_i}{\sum_{j=1}^{M} W_j} \rfloor \tag{6}$$

where $k_i$ is the number of frames allocated to subclip $C_i$.

Finally, for each subclip allocated multiple frames, we partition its temporal span into $k_i$ equal segments and select the frame with highest $S_{\text{text}}$ from each segment. This strategic partitioning prevents temporal clustering of selected frames while still maintaining focus on the most semantically

Table 1: Comparison of **training-free** token reduction/frame selection methods built on the LLaVA-OneVision model (LLaVA-OV) across benchmarks. CoSeLECT outperforms all methods in 3/4 tasks. (***Bold*** *denotes the best result per task,* underline *denotes the second-best result, and † indicates reproduced results.)*

| Method | Context Length | Frames pre-$E_{im}$ | VideoMME | MVBench | Long-VideoBench | Ego-Schema |
|---|---|---|---|---|---|---|
| LLaVA-OV (Baseline) | 8k | N/A | 58.4 | 57.8 | 56.8 | 62.8 |
| SlowFast-LLaVA (Xu et al., 2024c) | 3.6k | 32 | 56.1 | 56.4 | 55.1 | 61.4 |
| LLaVA-OV + Static or Dynamic (Shi et al., 2025b) | 8k | 128 | 59.9 | – | – | 60.5 |
| LLaVA-OV + QuoTA (Luo et al., 2025) | 8k | 64 | 60.7 | – | 57.4 | – |
| LLaVA-OV + BOLT (Liu et al., 2025) | 8k | fps | 59.9 | – | 59.6 | 64.0 |
| LLaVA-OV + AKS† (Tang et al., 2025b) | 8k | 64 | 58.2 | 57.4 | 56.4 | 62.6 |
| LLaVA-OV + AKS† (Tang et al., 2025b) | 8k | 400 | 60.2 | 56.6 | 57.4 | 61.4 |
| LLaVA-OV + AKS† (Tang et al., 2025b) | 8k | 1600 | **61.8** | 58.1 | 57.9 | 62.0 |
| LLaVA-OV + CoSeLECT (ours) | 8k | 32 | 57.7 | 57.8 | 57.1 | 63.4 |
| LLaVA-OV + CoSeLECT (ours) | 8k | 64 | 58.6 | 57.6 | 57.7 | **64.8** |
| LLaVA-OV + CoSeLECT (ours) | 8k | 400 | 59.7 | 58.1 | **59.3** | 64.0 |
| LLaVA-OV + CoSeLECT (ours) | 8k | 1600 | 61.1 | **58.2** | 58.8 | 63.0 |

relevant content within each temporal region. Without this partitioning, frame selection would likely cluster around peaks of relevance, potentially missing important context elsewhere in the subclip.

In summary, CoSeLECT provides a principled approach to query-aware frame selection by jointly modeling semantic relevance and visual continuity. This dual-signal strategy allows us to adaptively allocate limited frame tokens to the most informative and contextually important parts of a video. **We formally present our approach in Algorithm 1.**

## 4 EXPERIMENTS, RESULTS AND DISCUSSION

### 4.1 EXPERIMENTAL FRAMEWORK

**Implementation**  We use the image and text encoder from the finetuned version of SigLIP-ViT-SO400M-patch14-384 (Zhai et al., 2023) associated with the LLaVA-OneVision (Li et al., 2024a) to encode both video frames and the text query. Input images are resized to $384 \times 384$ and tokenized using a patch size of $14 \times 14$ before encoding. For the language foundation model, we report results using LLaVA-OneVision (Li et al., 2024a) with Qwen2-7B and Qwen2-0.5B as the decoder LLM. We also report results with CoSeLECT applied to Qwen2.5-VL-7B (Bai et al., 2025a) to demonstrate that our approach's effectiveness is agnostic to the choice of the base MLLM. We run all our evaluations on single 40G A40 nodes with 8 GPUs. We evaluate our method using varying sizes of the pre-embedding frame pool—specifically, 32, 64, 400, and 1600 frames—which we refer to as pre-$\mathbf{E}_{im}$ frames. Regardless of the pre-$\mathbf{E}_{im}$ pool size, we adaptively select 32 frames (unless specified otherwise), which we refer to as post-$\mathbf{E}_{im}$ frames before feeding image tokens into the LLM. This ensures that the sequence length of the tokens provided to the language model remains below 8k, allowing for consistent comparison across different frame selection strategies.

**Benchmarks**  We evaluate our models on well known video evaluation benchmarks: VideoMME(Fu et al., 2024a), MLVU(Zhou et al., 2025)'s `dev` split, MVBench(Li et al., 2024b), EgoSchema(Mangalam et al., 2023)'s `subset` split, LongVideoBench(Wu et al., 2024)' `val` set, NextQA(Xiao et al., 2021)'s `test` set.

### 4.2 KEY RESULTS

**Quantitative Comparison against Training-Free Methods**  Table 1 compares our training-free method with the LLaVA-OV baseline and other recent approaches under identical backbones. Since AKS is the current state of the art for training free adaptive frame selection, we not only reproduced their reported results by integrating their released code into our setup, but also extended their method to larger pre-$E_{im}$ budgets for a truly fair comparison. **Our method establishes a new state of the art in training-free frame selection, outperforming all competitors on 3 of 4 benchmarks and ranking second only on VideoMME.** The gains are especially pronounced on long-horizon

Table 2: Benchmarking performance of **trained** token reduction/frame selection methods against CoSeLECT, compared with training-based approaches. All methods use the same frame, LLM and vision encoder configurations within each block of the table (separated by midrules). *(Since the backbones aren't consistent, best performance is denoted per block. **Bold** denotes the best result within each block)*

| Method | Training Free | LLM Size | Vision Encoder Size | Context Length | Frames pre-$E_{im}$ | NextQA | MLVU |
|---|---|---|---|---|---|---|---|
| SeViLA (Yu et al., 2023) | ✗ | 3B | 1.1B | 2K | 32 | 73.8 | – |
| ViLA (Wang et al., 2024b) | ✗ | 3B | 1.4B | 2K | 32 | **74.8** | – |
| CoSeLECT + Qwen2.5-VL-3B-Instruct (Team, 2024) | ✓ | 3B | 0.4B | 2K | 32 | 74.6 | – |
| GCG (Wang et al., 2024a) | ✗ | 7B | 0.4B | 1K | 32 | 74.6 | – |
| CoSeLECT + LLaVA-OV | ✓ | 7B | 0.4B | 1K | 32 | **76.9** | – |
| Frame-Voyager (Yu et al., 2025) | ✗ | 7B | 0.4B | 2K | 32 | 73.9 | 65.6 |
| CoSeLECT + LLaVA-OV | ✓ | 7B | 0.4B | 2K | 32 | **78.0** | **66.1** |
| LongVA (Zhang et al., 2024) | ✗ | 7B | 0.4B | 224K | 32 | 69.3 | 56.3 |
| VideoChat2 (Li et al., 2024b) | ✗ | 7B | 0.3B | 8K | 16 | – | 47.9 |
| CoSeLECT + LLaVA-OV | ✓ | 7B | 0.4B | 8K | 32 | **80.0** | **63.5** |
| LongVU (Shen et al., 2024) | ✗ | 7B | 0.4B | 8K | 1fps | – | 65.4 |
| CoSeLECT + LLaVA-OV | ✓ | 7B | 0.4B | 8K | 1600 | **80.1** | **67.9** |

Table 3: Performance against token reduction techniques. Retained ratio indicates the percentage of original tokens remaining after token reduction. *(**Bold** denotes the best performance within each block.)*

| Method | Retained Ratio (%) | VideoMME | MVBench | MLVU | LongVideoBench | Average |
|---|---|---|---|---|---|---|
| LLaVA-OV | 100% | 58.4 | 57.8 | 62.4 | 56.8 | 58.9 |
| LLaVA-OV + Uniform frame sampling | 25% | 52.9 | 57.6 | 57.7 | 54.8 | 55.8 |
| LLaVA-OV + DyCoke Tao et al. (2025) | 25% | 51.0 | 49.5 | 55.8 | 48.1 | 51.1 |
| LLaVA-OV + FastV Chen et al. (2024b) | 25% | 56.2 | 54.7 | 61.5 | 55.5 | 57.0 |
| LLaVA-OV + VisionZip Yang et al. (2024) | 25% | 58.0 | 56.6 | 64.8 | 51.2 | 57.7 |
| LLaVA-OV + PruneVID Luo et al. (2025) | 25% | 57.5 | 55.0 | 64.6 | 55.4 | 58.1 |
| LLaVA-OV + FastVID Shen et al. (2025) | 25% | 58.0 | 56.5 | 64.1 | 56.3 | 58.7 |
| LLaVA-OV + CoSeLECT (Ours) | 25% | **58.0** | **57.9** | **66.1** | **57.4** | **59.9** |
| LLaVA-OV + Uniform frame sampling | 12.5%* | 50.1 | 55.8 | 55.6 | 50.9 | 53.1 |
| LLaVA-OV + FastV Chen et al. (2024b) | 15% | 54.7 | 53.2 | 59.8 | 54.9 | 55.7 |
| LLaVA-OV + VisionZip Yang et al. (2024) | 15% | 55.5 | 54.3 | 54.4 | 53.9 | 54.5 |
| LLaVA-OV + PruneVID Luo et al. (2025) | 15% | 56.1 | 54.6 | 63.1 | 53.6 | 56.9 |
| LLaVA-OV + FastVID Shen et al. (2025) | 15% | **57.7** | 56.0 | 63.3 | **56.2** | **58.3** |
| LLaVA-OV + CoSeLECT (Ours) | 12.5%* | 56.6 | **56.5** | **64.8** | 54.9 | 58.2 |

benchmarks: 64.8 on EgoSchema vs. the next best 62.8 (+2.0, a 3.2% relative gain), and 59.3 on Long-VideoBench vs. 57.9 (+1.4, a 2.4% relative gain).

**Quantitative Comparison against Trained Methods**   Table 2 compares our training-free method, CoSeLECT, against a range of trained frame-selection models. To ensure fairness, we align vision encoder, LLM, and pre-$E_{im}$ sizes, and report the best result in each cluster. Despite requiring no training, CoSeLECT outperforms most trained methods. Notably, many trained approaches only report results on small pre-$E_{im}$ budgets, as training becomes prohibitively expensive on larger frame sets. In contrast, our method remains scalable and competitive. Furthermore, models such as LongVU rely on computationally heavy spatiotemporal token compression, yet our simple similarity-based frame selection surpasses it (67.9 vs. 65.4 on MLVU). The only method ahead of us is ViLA, which leverages a much larger ViT-G vision encoder. *These results underscore that a lightweight, training-free strategy can rival—and in many cases surpass—specialized trained models.*

**Qualitative Results**   We qualitatively evaluate our model across diverse tasks and compare it to the baseline in Section A (Figures 3–6). Figure 7 illustrates how CoSeLECT adapts frame selection to the text query by showing differences in sampled frames for the same video under different questions.

Table 4: Exploration of performanceCoSeLECT across a variety of different LLM backbone varieties and sizes on a slew of tasks. CoSeLECT demonstrates consistent performance improvement across all backbones. pre-$E_{im}$ set to 800 and post-$E_{im}$ set to 32.

| Method | LLM | Context Length | VideoMME | MVBench | MLVU | LongVideo Bench | EgoSchema | Average |
|--------|-----|----------------|----------|---------|------|-----------------|-----------|---------|
| GPT-4o-mini | GPT-4o-mini | 8k | – | – | 64.9 | – | – | 64.9 |
| + CoSeLECT | GPT-4o-mini | 8k | – | – | 66.1 | – | – | **66.1** |
| LLaVA-OV | Qwen2-7B | 8k | 58.4 | 57.8 | 62.4 | 56.8 | 62.8 | 59.6 |
| + CoSeLECT | Qwen2-7B | 8k | 59.7 | 58.1 | 67.3 | 59.3 | 64.0 | **61.6** |
| LLaVA-OV | Qwen2-0.5B | 8k | 43.7 | 46.3 | 46.5 | 46.8 | 26.6 | 42.0 |
| + CoSeLECT | Qwen2-0.5B | 8k | 45.6 | 46.5 | 49.5 | 47.8 | 26.6 | **43.2** |
| Qwen2.5-VL-7B-Inst | Qwen2.5-7B | 8k | 61.3 | 68.3 | 59.5 | 58.9 | 59.8 | 61.6 |
| + CoSeLECT | Qwen2.5-7B | 8k | 63.3 | 69.5 | 63.4 | 59.2 | 59.6 | **63.0** |

Table 5: Exploring optimal frame selection strategy. Demonstrating why our method incorporates **both** text similarity and visual continuity signal—not just one or the other.

| Frame selection method | Frames pre-$E_{im}$ | Frames post-$E_{im}$ | VideoMME | MVBench | MLVU | Long-VideoBench | EgoSchema | Average |
|------------------------|---------------------|----------------------|----------|---------|------|-----------------|-----------|---------|
| Uniform sampling | 32 | 32 | 58.4 | 57.8 | 62.4 | 56.8 | 62.8 | 59.6 |
| 32 top text similarity frames | 400 | 32 | 60.7 | 57.6 | 65.7 | 57.7 | 61.8 | 60.7 |
| Only Visual Continuity | 400 | 32 | 59.1 | 57.9 | 64.9 | 57.6 | 63.0 | 60.5 |
| CoSeLECT (Ours) | 400 | 32 | 59.7 | 58.1 | 65.9 | 59.3 | 64.0 | **61.4** |

## 4.3 ABLATIONS

**Efficient Video Understanding** Table 3 evaluates performance in a token-constrained scenario—passing only a small number of tokens to the downstream LLM—by benchmarking our method against state-of-the-art, training-free token pruning techniques. While prior methods perform pruning directly over tokens, our approach reduces tokens indirectly by selecting a subset of informative frames. Thus, it can be viewed as a frame-level token reduction strategy. Specifically, a 25% pruning ratio corresponds to selecting 8 of the original 32 frame tokens, and 12.5% pruning uses just 4. We begin with 400 frames per video and adaptively select the top frames to match these targets. Our method consistently matches or exceeds state-of-the-art results under extreme compression. At 25%, we achieve the highest VideoMME score, tying with VisionZip and FastVID. At 12.5%, we outperform all competitors on MVBench and MLVU—even with fewer input frames. These results suggest that *high-quality token reduction can be achieved by frame selection alone, without operating at token granularity, when leveraging a rich initial candidate pool.*

**Robustness across Architectural Changes** We demonstrate CoSeLECT's robustness across model architectures and scales (Table 4). In our baseline setting, when applied to LLaVA-OV with Qwen2-7B, CoSeLECT improves the average score from 59.6 to 61.6 (3.4% relative gain). These gains persist when scaling down the LLM size: with the smaller Qwen2-0.5B backbone, CoSeLECT boosts performance from 42.0 to 43.2 (2.9% relative gain). The effectiveness of our approach also extends beyond a single MLLM family. On Qwen2.5-VL-7B-Inst, CoSeLECT improves average performance from 61.6 to 63.0 (2.3% relative gain). Finally, we observe consistent gains even on proprietary closed-source models: applied to GPT-4o-mini, CoSeLECT increases performance from 64.9 to 66.1 (1.8% relative gain). *Together, these results highlight the robustness and generality of CoSeLECT across diverse architectures and scales.*

**Complementary Signals for Frame Selection** Our adaptive strategy selects frames that are *both semantically aligned with the query and temporally diverse.* As shown in Table 5, CoSeLECT outperforms query-relevance-only selection across all benchmarks, with an average gain of 0.9%. Notably, the visual continuity signal alone already improves over uniform sampling ($59.6 \rightarrow 60.5$) and performs on par with text similarity (60.7 vs. 60.5). Together, they yield the best overall score of 61.4. *When combined, the two signals complement each other: semantic relevance guides selection toward informative frames, while continuity ensures diverse coverage and provides a reliable fallback that consistently outperforms uniform sampling.*

Table 6: Performance comparison across encoder configurations in the similarity computation phase. The backbone is LLaVA-OneVision-7B with pre-$E_{im}$ set to 800 and post-$E_{im}$ set to 32. The first two columns specify the encoders used for Image–Image similarity (Visual Continuity Signal) and Image–Text similarity (Text Relevance Signal). (**Bold** *denotes the best average result.*)

| Image–Image Sim | Image–Text Sim | Num Params | ViMME | MVBench | MLVU | LongVideo Bench | EgoSchema | Average |
|---|---|---|---|---|---|---|---|---|
| CLIP-B | CLIP-B | 151M | 59.9 | 58.3 | 66.3 | 58.5 | 63.6 | 61.3 |
| DINO-ViT-B/16 | SIGLIP | 86M + 878M | 60.8 | 58.3 | 67.3 | 59.3 | 63.6 | **61.9** |
| SIGLIP | SIGLIP | 878M | 60.6 | 58.3 | 66.2 | 59.3 | **64.5** | 61.8 |

**Impact of encoder choice**   As shown in Table 6, lightweight encoders such as CLIP-B achieve competitive performance across benchmarks, with only minor degradation relative to larger models like SIGLIP or DINO-ViT-B/16. We believe this serves as a practical datapoint for users selecting configurations under compute, time, or task constraints.

**Dissecting CoSeLECT further**   Beyond the core ablations, Appendix B presents a more exhaustive exploration of CoSeLECT's mechanics and hyperparameters. We analyze the composite relevance and duration weighting schemes, the effect of initial frame pool size, and sensitivity to the visual similarity threshold for scene segmentation. These results provide deeper justification for our design choices and further highlight CoSeLECT's advantages over alternatives.

## 4.4 ON PERFORMANCE OVERHEAD OF COSELECT

In our algorithm, we encode $N$ (where $32 \leq N \leq 1600$) frames per data sample using the So400M-patch14-384 SigLIP Encoder (Zhai et al., 2023). While this might appear to introduce substantial overhead, dense frame sampling is becoming increasingly common in video-centric multimodal LLMs—e.g., Apollo (Zohar et al., 2024) and LongVU (Shen et al., 2024)—which both advocate for and benefit from fps sampling. *Our approach thus aligns with this emerging trend, rather than representing an atypical design choice.*

Moreover, the independent nature of each forward pass renders the process embarrassingly parallel and well-suited for batching or distributed inference. As demonstrated in Figure 2, FLOPs scale linearly with $N$ as expected, but practical latency remains significantly lower than what the FLOP count would suggest. For example, on an 8-GPU A40 node with 40 GB memory, encoding 1600 frames requires $32\times$ the FLOPs of encoding 50 frames, yet incurs only a $4.1\times$ latency increase. We also plot the raw latency values in C.1 for reference. *This gap highlights that, under parallel execution conditions, the wall-time burden of our algorithm is markedly less severe than its theoretical complexity implies.*

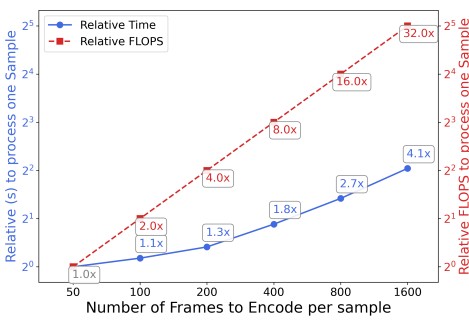

Figure 2: Relative FLOPs and wall-clock latency as we increase frame counts

## 5 CONCLUSION

In this work, we introduce CoSeLECT, a training-free adaptive frame selection approach addressing the fundamental challenge of efficiently processing long videos within the token constraints of current MLLMs. By effectively balancing temporal redundancy and semantic relevance, CoSeLECT dynamically adapts frame selection to both video content and the input query, overcoming critical limitations of naive sampling and costly fine-tuning methods. Extensive experiments across diverse benchmarks demonstrate that CoSeLECT consistently matches or outperforms state-of-the-art approaches—including fine-tuned and dynamic token pruning baselines—while maintaining strong generalization across multiple MLLM architectures and scales. Given its simplicity and effectiveness, we hope that our approach will be adopted, extended, and leveraged across a broad spectrum of future multimodal models and applications.

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

## A APPENDIX: QUALITATIVE RESULTS

We now present qualitative comparisons between CoSeLECT and the baseline LLaVA-OneVision Li et al. (2024a) across a diverse set of task types. These examples highlight how our frame selection strategy leads to more informative context for downstream video understanding.

In Figure 3, we consider a "needle in a haystack" setting, where the correct answer depends on identifying a single rare frame; CoSeLECT successfully surfaces the key frame while uniform sampling fails. Figure 4 examines plot-level reasoning, where CoSeLECT selects a broader variety of frames documenting different activities, including the critical one needed to answer the query. In Figure 5, we show an egocentric scenario where a subtle visual cue (a red clothing artifact) is crucial—CoSeLECT captures it, enabling correct reasoning. Figure 6 demonstrates topic-level reasoning, with our method surfacing semantically diverse frames (e.g., beach scenes, palm trees) that enrich context. Finally, Figure 7 illustrates the query-aware nature of our approach: when posed with different queries on the same long video, CoSeLECT adapts its selections to highlight distinct but relevant evidence, minimizing redundancy while maintaining diversity. Together, these case studies demonstrate that CoSeLECT consistently selects frames that are both query-relevant and temporally diverse, thereby providing richer context than uniform or text-only sampling.

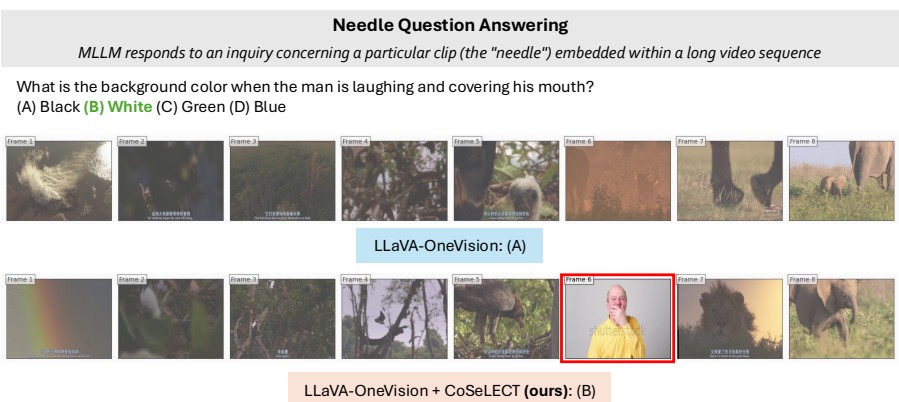

Figure 3: **Needle Question Answering:** CoSeLECT successfully localizes the "needle" to provide an accurate answer, whereas uniform sampling fails to do so.

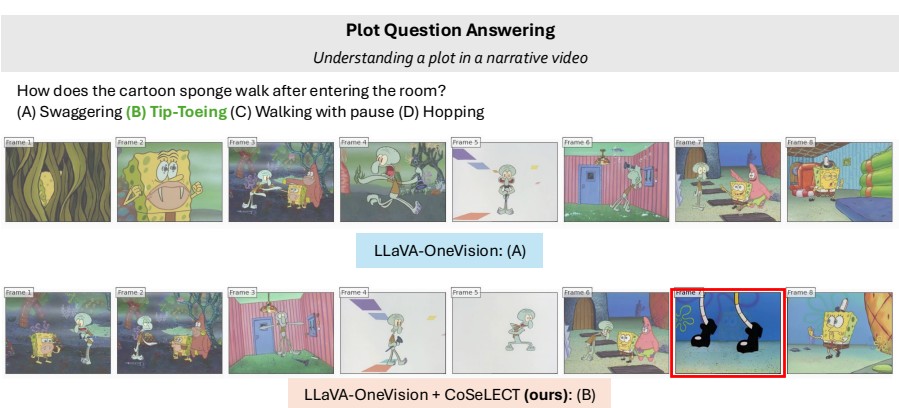

Figure 4: **Plot Question Answering:** CoSeLECT samples diverse frames showing different types of walking, including the one containing the correct answer.

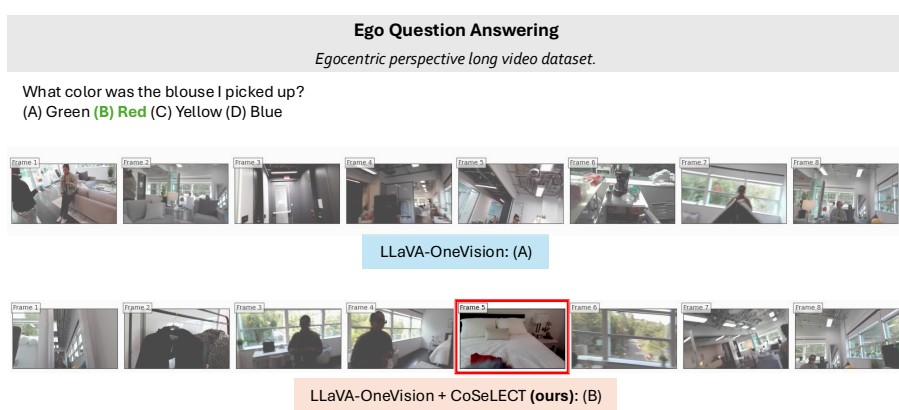

Figure 5: **Ego Question Answering:** CoSeLECT captures a subtle artifact of red clothing in the frame—providing just enough context for the LLM to answer correctly.

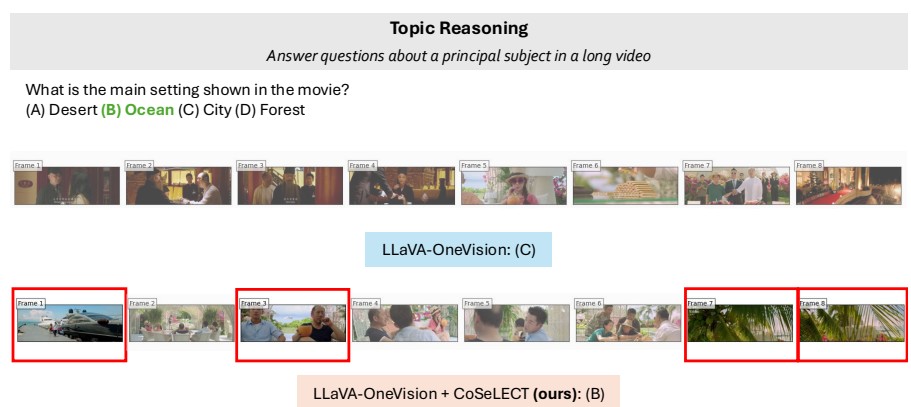

Figure 6: **Topic Reasoning:** CoSeLECT captures a broader diversity of concepts related to the query (e.g., multiple scenes by the beach, palm trees), providing richer context for the LLM.

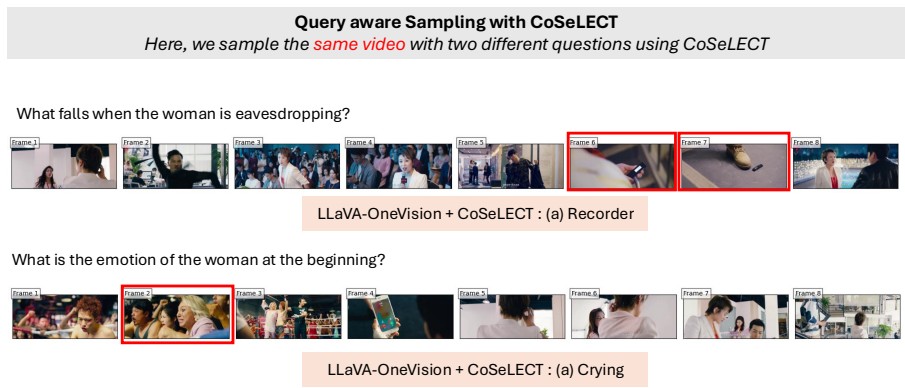

Figure 7: **Query-aware sampling in action:** CoSeLECT selects different frames for different queries from the same long source video. Two observations stand out: (1) In both cases, it surfaces relevant frames and produces the correct answer; (2) The selected frames exhibit minimal redundancy, showing that CoSeLECT generates relevant yet diverse frames.

# B  APPENDIX: DISSECTING THE METHOD - ABLATIONS AND INSIGHTS

## B.1  FORMAL ALGORITHM

We provide the formalized algorithm of the method described in 3 below.

---

**Algorithm 1** CoSeLECT- Query-Aware Frame Selection

---

**Require:** Video $V = \{f_1, f_2, ..., f_N\}$, query $Q$, image encoder $E_{im}$, text encoder $E_{text}$, output frames $K$

   // —- Embedding Extraction —-

   Extract embeddings $E_{im}(f_i)$ for all frames and $E_{text}(Q)$ for query

   // —- Text Relevance Signal —-

   $S_{\text{text}}(t) = \text{smooth}(E_{im}(f_t) \odot E_{text}(Q))$             # Compute query-frame relevance

   // —- Visual Continuity Signal —-

   $S_{\text{frame}}(t) = \text{smooth}(E_{im}(f_t) \odot E_{im}(f_{t+1}))$           # Compute frame-to-frame similarity

   $\text{SceneBoundaries} = \{k \mid S_{\text{frame}}(k) < \tau_{\text{sim}}\}$             # $\tau_{\text{sim}} = 0.8$

   Partition video into subclips $C = \{C_1, ..., C_M\}$ at scene boundaries

   Record duration $D_i$ for each subclip $C_i$

   // —- Compute Composite Relevance —-

   **for** each $C_i$ in $C$ **do**

      $R_i = \max(S_{\text{text}}|_{C_i}) + \text{mean}(S_{\text{text}}|_{C_i})$

   **end for**

   // —- Adaptive Frame Selection —-

   **for** each $C_i$ in $C$ **do**

      $W_i = R_i \cdot \sqrt{D_i}$             # Duration-adjusted relevance weight

   **end for**

   $k_i = \lfloor K \cdot \frac{W_i}{\sum_{j=1}^{M} W_j} \rfloor$             # Distribute frame budget

   **for** each $C_i$ with allocation $k_i > 0$ **do**

      Partition $C_i$ into $k_i$ equal temporal segments

      Select frame with highest $S_{\text{text}}$ from each segment

   **end for**

   **return** $K$ Selected Frames

---

## B.2  EFFECT OF COMPOSITE RELEVANCE AND DURATION WEIGHTING

Table 7: Ablation of key design choices in CoSeLECT, corresponding to Algorithm 1. We test variants that remove or simplify components of the composite relevance and weighting scheme.

| Variant | VideoMME | MVBench | MLVU | Long-VideoBench | EgoSchema | Average |
|---|---|---|---|---|---|---|
| $D_i$ instead of $\sqrt{D_i}$ | 60.1 | 58.1 | 66.2 | 58.6 | 63.6 | 61.3 |
| $R_i = \max(S_{\text{text}}|_{C_i})$ only | 60.1 | 57.7 | 66.0 | 58.0 | 63.6 | 61.1 |
| $R_i = \text{mean}(S_{\text{text}}|_{C_i})$ only | 60.5 | 58.1 | 66.0 | 58.8 | 63.8 | 61.4 |
| **CoSeLECT** | **60.6** | **58.3** | **66.2** | **59.3** | **64.5** | **61.8** |

Table 7 highlights the contribution of each component in our design. We find that replacing the duration adjustment term $\sqrt{D_i}$ with a linear factor $D_i$, or simplifying the composite relevance to rely solely on either the maximum or the average similarity score in $R_i = \max(S_{\text{text}}|_{C_i}) + \text{mean}(S_{\text{text}}|_{C_i})$, consistently lowers performance. In contrast, combining both max and average signals with square-root duration weighting yields the strongest results across benchmarks.

## B.3 CHOOSING SIMILARITY THRESHOLD

Table 8: Impact of frame-to-frame similarity threshold $\tau_{sim}$ on CoSeLECT performance across benchmarks. This threshold corresponds to the parameter in Algorithm 1, where scene boundaries are defined as $\{k \mid S_{frame}(k) < \tau_{sim}\}$ (e.g., $\tau_{sim} = 0.8$).

| Threshold | VideoMME | MVBench | MLVU | Long-VideoBench | EgoSchema | Average |
|-----------|----------|---------|------|-----------------|-----------|---------|
| 0.2 | 60.0 | 58.2 | 67.1 | 58.4 | 64.2 | 61.6 |
| 0.4 | 60.0 | 58.0 | 66.8 | 57.3 | 63.8 | 61.2 |
| 0.6 | 60.0 | 57.9 | 66.8 | 58.0 | 63.8 | 61.3 |
| 0.8 | 60.1 | 58.1 | 67.3 | 59.3 | 64.0 | **61.8** |

From our test set we obtain the best performance at $\tau_{sim} = 0.8$. This parameter can further be tuned for any new tasks as well.

## B.4 EFFECT OF VARYING THE PRE-$E_{im}$

Table 9: Effect of varying the pre-$E_{im}$ frame pool size on CoSeLECT performance across benchmarks. Here, pre-$E_{im}$ denotes the number of candidate frames encoded before applying CoSeLECT, while post-$E_{im}$ indicates the final number of selected frames passed to the LLM.

| Method | Frames pre-$E_{im}$ | Frames post-$E_{im}$ | VideoMME | MVBench | MLVU | LongVideoBench | Average |
|--------|---------------------|----------------------|----------|---------|------|----------------|---------|
| LLaVA-OV + Uniform Sampling | 32 | 32 | 58.4 | 57.8 | 62.4 | 56.8 | 58.9 |
| LLaVA-OV + CoSeLECT | 32 | 32 | 57.7 | 57.8 | 63.5 | 57.1 | 59.0 |
| LLaVA-OV + CoSeLECT | 64 | 32 | 58.6 | 57.6 | 65.1 | 57.7 | 59.8 |
| LLaVA-OV + CoSeLECT | 100 | 32 | 59.0 | 57.9 | 64.5 | 56.5 | 59.5 |
| LLaVA-OV + CoSeLECT | 200 | 32 | 58.9 | 57.8 | 65.2 | 58.3 | 60.1 |
| LLaVA-OV + CoSeLECT | 400 | 32 | 59.7 | 58.1 | 67.3 | **59.3** | 61.1 |
| LLaVA-OV + CoSeLECT | 800 | 32 | 60.6 | **58.3** | 66.2 | **59.3** | 61.1 |
| LLaVA-OV + CoSeLECT | 1600 | 32 | **61.1** | 58.1 | **67.9** | 58.8 | **61.5** |

pre-$E_{im}$ refers to the number of encoded frames that the selection algorithm runs on to make its selection. Table 9 highlights how the size of the pre-embedding frame pool impacts performance. Increasing the candidate pool from 32 frames (uniform sampling) to 400 frames (our method) improves average performance from 58.9 to 61.1, with gains particularly notable on tasks like VideoMME (from 58.4 to 59.7, +1.3) and Long-VideoBench (from 56.8 to 59.3, +2.5). However, this trend is not uniform across benchmarks—MVBench shows only marginal improvements (+0.3), suggesting that the optimal pool size is task-dependent. We also observe diminishing returns: for instance, switching from uniform sampling (32 frames) to top text-similarity frames (400 frames) yields a +2.2 average gain, while further increases yield smaller improvements. These findings suggest that tuning the number of candidate frames using a validation set is beneficial. Furthermore, this parameter offers a flexible trade-off between performance and compute, allowing our method to adapt to different resource budgets—a practical strength in real-world settings.

## B.5 EFFECT OF VARYING THE POST-$E_{im}$

Table 10: Effect of increasing the number of post-frames on baseline uniform sampling and CoSe-LECT. Pre-embedding pool size is fixed at 1600.

| Method | Post-Frames | VideoMME | MVBench | MLVU | Long-VideoBench | EgoSchema | Average |
|--------|-------------|----------|---------|------|-----------------|-----------|---------|
| Baseline | 32 | 58.4 | 57.8 | 62.4 | 56.8 | 62.8 | 59.6 |
| **CoSeLECT** | 32 | **61.1** | **58.1** | **67.9** | **58.8** | **63.0** | **61.8** |
| Baseline | 64 | 58.7 | 56.9 | 64.3 | 57.3 | 63.0 | 60.0 |
| **CoSeLECT** | 64 | **59.7** | **57.1** | **66.8** | **57.8** | 62.4 | **60.8** |
| Baseline | 128 | 58.3 | 56.6 | 65.9 | 56.9 | 63.4 | 60.2 |
| **CoSeLECT** | 128 | **58.9** | 56.8 | **66.3** | **57.7** | **63.6** | **60.7** |

It may seem intuitive that passing more frames into the downstream LLM should improve performance. However, as shown in Table 10, increasing the number of post-$E_{\text{im}}$ frames beyond 32 does not consistently lead to gains. We attribute this to two factors: (1) **Redundancy** — longer frame sequences often contain many visually similar or irrelevant frames, reducing the signal-to-noise ratio. (2) **Model limitations** — current MLLMs have bounded attention capacity, so adding more frames can dilute the contribution of the most informative ones.

Notably, CoSeLECT maintains relatively higher performance across all frame settings, suggesting that adaptive frame selection mitigates both redundancy and attention bottlenecks.

### B.6 COMPARISON WITH ADAPTIVE KEYFRAME SAMPLING (AKS)

Adaptive Keyframe Sampling (AKS) (Tang et al., 2025a) is a recent training-free method that, like CoSeLECT, leverages frame–text similarity for plug-and-play frame selection. While the two share a high-level motivation, they differ fundamentally in how they achieve coverage and allocate frames. Below, we outline the AKS algorithms explicitly and then provide a side-by-side comparison.

---

**Algorithm 2** AKS Algorithm

---

1: **Segmentation:** Recursively split the video into equal halves.
2: **if** (max text-sim – avg text-sim) > threshold **or** recursion depth reached **then**
3:     stop splitting
4: **else**
5:     recurse on both halves
6: **end if**
7: **Budget Allocation:** Assign frame budget inversely proportional to recursion depth.
8: **Frame Selection:** Select top-k most text-relevant frames in each segment. Every segment receives at least one frame.

---

**Comparison.** Despite their different designs, both AKS and CoSeLECT share the same underlying goal: to select a subset of frames $S$ that are both highly relevant to the query and sufficiently representative of the video as a whole. This trade-off can be abstracted as optimizing

$$\arg \max_{S \subset \text{frames}} \left[ \alpha \cdot \text{Relevance}(S, \text{query}) + \beta \cdot \text{Coverage}(S, \text{video}) \right].$$

The two methods adopt a similar approach for computing **relevance** by leveraging frame–text similarity, but they diverge fundamentally in how they define and enforce **coverage**. AKS relies on statistical variance and recursive partitioning, while CoSeLECT incorporates semantic continuity through inter-frame similarity and adaptive budgeting. Table 11 summarizes these distinctions in detail.

Table 11: Algorithmic comparison of AKS and CoSeLECT.

|  | **Adaptive Keyframe Sampling (AKS)** | **CoSeLECT (Ours)** |
|---|---|---|
| **Segmentation** | **Statistical, top-down:** recursively splits the video into equal halves based on text-similarity variance. Segmentation is agnostic to scene boundaries. | **Semantic, bottom-up:** segments the video into variable-length, coherent subclips using inter-frame similarity, aligning with scene changes. |
| **Budget Allocation** | **Indirect proxy:** frame budget is determined by recursion depth; deeper segments receive fewer frames. | **Direct signals:** frame budget is allocated using subclip length and average text–frame similarity. |
| **Frame Selection** | **Forced allocation:** selects top-k text-relevant frames per segment; every segment must contribute at least one frame. | **Flexible allocation:** selects top-k text-relevant frames; irrelevant subclips can be skipped entirely (zero budget). |

Table 12: Performance on three benchmarks with Qwen2.5-VL-7B as the backbone. All methods use pre-$E_{\text{im}} = 800$ and post-$E_{\text{im}} = 32$.

| Model | VideoMME | MLVU | LVBench |
|---|---|---|---|
| Qwen2.5-VL-7B + AKS | 63.2 | 61.4 | 58.9 |
| Qwen2.5-VL-7B + Q-Frame | 58.3 | **65.4** | 58.4 |
| Qwen2.5-VL-7B + CoSeLECT | **63.3** | 63.4 | **59.2** |

Table 13: Control experiment with Qwen2.5-VL under a fixed token-per-frame budget (max_pixels = 90k, min_pixels = 80k). The baseline uses 256 uniformly sampled frames; CoSeLECT selects 256 frames from a 1600-frame pool under the same context constraints.

| Setting | VideoMME w/o subs | MLVU |
|---|---|---|
| Reported baseline | 65.1 | 71.6 |
| Reproduced baseline (uniform frames) | 64.8 | 71.1 |
| + CoSeLECT (1600→256 frames) | **66.2** | **71.9** |

This makes clear that CoSeLECT differs from AKS in two central respects: (1) **Semantic vs. Statistical Segmentation:** CoSeLECT explicitly respects inter-frame continuity, while AKS partitions by statistical thresholds over similarity scores. (2) **Dual-signal vs. Single-signal Coverage:** CoSeLECT combines frame–frame and frame–text signals for coverage and relevance, whereas AKS relies solely on variance in frame–text similarity.

### B.7 COMPARISON ACROSS FRAME SELECTION METHODS WITH A DIFFERENT BACKBONE

Here we add an additional ablation in 12 that shows performance across AKSTang et al. (2025a) and Q-FrameZhang et al. (2025) with a different vision backbone.

### B.8 COMPARISON ACROSS LARGER DOWNSTREAM FRAME COUNTS

To isolate the effect of frame selection under a fixed context budget, we compare uniform sampling with CoSeLECT when selecting 256 frames. As shown in Table 13, CoSeLECT consistently improves over the uniform baseline on both MLVU and VideoMME, indicating that the gains arise from *which* frames are selected rather than from changes in the token-per-frame setting.

# C APPENDIX: ADDITIONAL RESULTS

## C.1 COSELECT'S EFFECT ON WALL-TIME LATENCY

Figure 8 visualizes how increasing the size of the pre-embedding frame pool impacts wall-time when running our method on a 40 GB A40 node with 8 GPUs.

It is worth noting that the majority of the time is spent in the vision encoding and LLM forward pass, with only $0.08\%$ of the time spent on inter-frame and text–frame embedding comparisons.

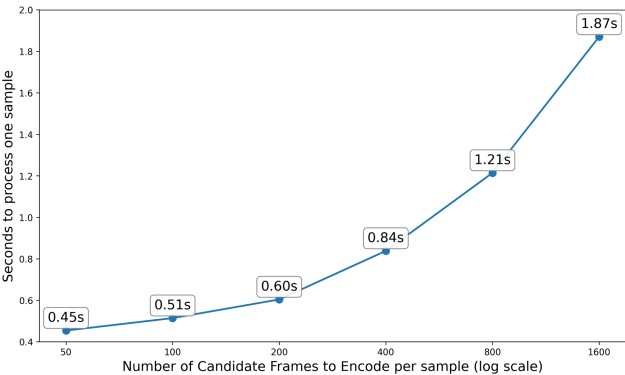

Figure 8: This chart shows the empirically measured inference latency as the size of the pre-embedding frame pool is varied.

## C.2 DETAILED LONGVIDEOBENCH RESULTS

Table 14: Per-subtask performance on Long-VideoBench.

| Method | TOS | S2E | E3E | S2A | SAA | O3O | T3O | T3E | O2E | T2O | S2O | TAA | T2E | E2O | SSS | T2A | SOS |
|---|---|---|---|---|---|---|---|---|---|---|---|---|---|---|---|---|---|
| Baseline (%) | 34.2 | 68.8 | 70.2 | 76.1 | 54.2 | 53.0 | 52.7 | 49.3 | 62.1 | 57.9 | 52.8 | 47.6 | 58.5 | 67.7 | 36.1 | 59.5 | 61.7 |
| CoSeLECT 1600 pre (%) | 35.6 | 72.0 | 63.8 | 80.7 | 58.3 | 54.5 | 51.4 | 49.3 | 72.4 | 61.8 | 56.9 | 56.1 | 61.5 | 67.7 | 32.0 | 59.5 | 63.0 |
| CoSeLECT 64 pre (%) | 32.9 | 67.7 | 63.8 | 71.6 | 56.9 | 48.5 | 55.4 | 49.3 | 69.0 | 63.2 | 55.6 | 50.0 | 53.8 | 64.6 | 33.0 | 55.7 | 65.4 |

# D APPENDIX: LIMITATIONS

CoSeLECT inherits the biases and failure modes of the underlying vision–language encoders (e.g., SigLIP, Qwen2.x). In addition, our method currently operates on RGB frames only and does not exploit audio or motion-specific features, which may be important for certain tasks.

# E APPENDIX: LLM USAGE

We used large language models (LLMs) such as ChatGPT to assist with proofreading and polishing the presentation of this paper. Their role was limited to improving readability, grammar, and flow; all technical content, experimental design, and scientific claims were developed, implemented, and validated by the authors. The use of LLMs was therefore restricted to editorial support, without influence on the novelty or substance of the research.

