# OpenReview forum: "Training-Free Adaptive Frame Selection for Video-Language Understanding"
_ICLR.cc/2026/Conference — Submitted to ICLR 2026_

### Official Review · Reviewer_Lg9P · 2025-10-22

**Soundness:** 2
**Presentation:** 2
**Contribution:** 2
**Rating:** 4
**Confidence:** 5

**Summary:**

The paper proposes CoSeLECT, a training-free adaptive frame selection method that efficiently selects the most informative frames from a large pool by combining query relevance and temporal continuity. This approach achieves better performance than existing training-free methods on multiple video understanding benchmarks.

**Strengths:**

The paper presents a clear and practical training-free frame selection method that combines query relevance and visual continuity in a straightforward manner. While the individual components are not novel, their combination into an adaptive, query-aware selection pipeline is sensibly designed and effectively executed. The method is well-described, easy to reproduce, and evaluated across multiple benchmarks with ablations that support key design choices. Its significance lies in offering a lightweight, plug-and-play solution that improves efficiency and performance for video understanding with MLLMs without requiring model retraining. This is useful for real-world deployment, though not theoretically groundbreaking.

**Weaknesses:**

- The novelty of the paper is limited. Query-aware frame selection for Videl-LLM is not innovative, as discussed in KeyVideoLLM [1], AKS [2], and Q-Frame [3]. The paper pointed out that `these heavier methods are typically limited to sparsely pre-sampled frame pools` is interesting, but the experiment did not support the solution of this problem.

- It is not clear whether the comparison of the experimental results in the paper with other training-based methods is fair.

- The paper claims the proposed CoSeLECT is lightweight. However, there is a lack of systematic evaluation of latency and computing consumption, which is crucial for actual deployment.
> but in its lightweight, principled fusion of two readily available ones—frame–text similarity for semantic relevance and inter-frame similarity for temporal continuity.

- Lack of discussion of limitations.

- Minor weaknesses
  - The first equation in Section 3.4 is missing a number
  - $\sqrt{D_i}$ in equation (4) lacks definition

[1] Liang H, Li J, Bai T, et al. Keyvideollm: Towards large-scale video keyframe selection[J]. arXiv preprint arXiv:2407.03104, 2024.
[2] Tang X, Qiu J, Xie L, et al. Adaptive keyframe sampling for long video understanding[C]//Proceedings of the Computer Vision and Pattern Recognition Conference. 2025: 29118-29128.
[3] Zhang S, Yang J, Yin J, et al. Q-Frame: Query-aware Frame Selection and Multi-Resolution Adaptation for Video-LLMs[J]. arXiv preprint arXiv:2506.22139, 2025.

**Questions:**

- I am confused about the avoidance of redundant calculations mentioned in the article in line 204. Although  LLaVA-OneVision uses SigLIP-So400M-patch14-384 as the visual encoder, it fine-tunes SigLIP during the training process, which results in them having the same structure but different parameters. So is it really possible to avoid redundant calculations?
> Since SigLIP is also used as the vision encoder in LLaVA-OneVision (Li et al., 2024a), these embeddings can be directly reused, avoiding redundant computation.

- Does pre-$\textbf{E}_{im}$  in section 4 mean $N$ in section 3, and post-$\textbf{E}_{im}$  means  $K$? If so, please use consistent expressions to improve reading; if not, I hope the author can further elaborate on the difference.

- Lack of in-depth analysis of Table 1. With the increase of pre-$\textbf{E}_{im}$, there is no consistent performance improvement across all benchmarks. Does this contradict the motivation of the paper?
> Crucially, these heavier methods are typically limited to sparsely pre-sampled frame pools in order to remain computationally feasible—risking the permanent loss of “needle-in-a-haystack” moments before the selection algorithm can even evaluate them, a limitation that becomes particularly acute under resource constraints.

- I am confused about the experimental results in Table 2. Is this comparison meaningful?
  - Frame-Voyager is similar to the proposed CoSeLECT, and is also a plug-and-play model. But its LLM Size does not seem to be 7B.
  - LongVA and VideoChat2 are Video-LLMs. How to compare them with CoSeLECT ?

- Supplement CoSeLECT compares the results of the Qwen2-VL [1] experiment with AKS and Q-Frame [2], which will provide a more comprehensive evaluation.

[1] Wang P, Bai S, Tan S, et al. Qwen2-vl: Enhancing vision-language model's perception of the world at any resolution[J]. arXiv preprint arXiv:2409.12191, 2024.

[2] Zhang S, Yang J, Yin J, et al. Q-Frame: Query-aware Frame Selection and Multi-Resolution Adaptation for Video-LLMs[J]. arXiv preprint arXiv:2506.22139, 2025.

---

> ### Author Response · Authors · 2025-11-21
> **Thank you for the feedback!**
>
> We are extremely grateful to the reviewer for their thoughtful reviews and valuable suggestions!
>
> > pre-$E_{\text{im}}$ vs performance (The paper pointed out ... this problem & Lack of in-depth analysis ... motivation of the paper?)
>
> Thank you for raising this point. Our motivation is *not* that performance must increase monotonically with pre-$E_{\text{im}}$, but that having access to a *denser pool* of candidate frames and an adaptive selector is preferable to committing to a very sparse, fixed pre-sampling strategy. Table 1 and Appendix B.4 show exactly this: moving from 32 to 400 candidate frames yields clear average gains and especially helps long-horizon tasks, while very large pools show task-dependent saturation and mild non-monotonicity. In contrast, many SOTA systems such as LongVU [1] and AKS [2] effectively operate with fixed per-frame sampling and always encode all sampled frames. Our results indicate that the "sweet spot" in pre-$E_{\text{im}}$ is **task-dependent**, and CoSeLECT exposes pre-$E_{\text{im}}$ as a practical compute–accuracy knob so users can allocate inference-time resources more carefully rather than always encoding every frame or only 32 frames.
>
>
> > Table 2 clarity (I am confused ... meaningful? & It is not clear .... methods is fair.)
>
> 1) For **Frame-Voyager** [6], we use the numbers from Table 1 (third block) of Yu et al. (2024), where Frame-Voyager is used with **LLaVA-OneVision-7B**—the same backbone size and vision encoder as in our setup.
> 2) Both **LongVA** [5] and **VideoChat2** [4] are end-to-end trained Video-LLMs, while CoSeLECT is a training-free selector applied on top of a pretrained multimodal LLM. For these trained works in Table 2, we use the authors' reported results, and ensure the LLM and vision parameter counts are comparable to ours even though the architectures differ (Video-LLM vs. LLaVA-OV/Qwen + CoSeLECT). Following prior state-of-the-art works (e.g., LongVU [1]), which also report against these models on the same benchmarks, we treat them as **reference Video-LLM baselines at similar scale**.
> **This allows Table 2 to indicate where a training-free CoSeLECT + LLaVA/Qwen system sits relative to commonly used end-to-end trained Video-LLMs.**
>
>
> > Systematic evaluation of latency and computing consumption (The paper claims the proposed CoSeLECT is lightweight .... temporal continuity.)
>
> We provide the raw latency measurements in Appendix C.1. The dominant costs in our pipeline are similar to fps-sampling methods such as AKS [2], Q-Frame [3], and LongVU [1]: encoding frames with the vision encoder and the subsequent LLM forward pass. CoSeLECT adds only an $\mathcal{O}(N)$ similarity computation over the pre-embedded frames (pairwise vision–vision and vision–text cosine similarities). In practice, this overhead is negligible: for pre-$E_{\text{im}} = 1600$ (the worst case), the similarity computation takes **1.49 ms** out of **1.87 s** total inference time (≈**0.08%**) per query.
>
> **Changes:** We have explicitly highlighted these measurements in the Appendix C.1.
>
>
> > Avoidance of redundant calculations (I am confused .... redundant calculations?)
>
> The reviewer is correct. Our claim applies specifically to our evaluation setting: our current setting uses the **same** visual encoder for CoSELECT scoring as is used in the base MLLM for visual encoding. We encode each frame once, cache $E_{\text{im}}(f_t)$, and reuse it. This optimization would not apply if a different vision encoder were used downstream.
>
> **Changes:** We have clarified this scope and softened the wording in Sec. 3.2.
>
>
> > Qwen2-VL + AKS and Q-Frame (Supplement CoSeLECT .... comprehensive evaluation.)
>
> **Changes:** We have now added this comparison table to the paper, see appendix B.7
>
>
> > Misc Corrections
>
> Thank you for pointing out these corrections!
>
> **Changes:**
> * We have revised the paper to explicitly state that pre-$E_{\text{im}}$ = $N$ and post-$E_{\text{im}}$ = $K$ in section 3.2.
> * We have changed the paper to fix the **minor weaknesses** - equation corrections.
> * We have added a **limitations** section in the appendix.
>
> > References
>
> [1] Shen X, et al. LongVU: Spatiotemporal Adaptive Compression for Long Video-Language Understanding. *arXiv preprint arXiv:2410.17434*, 2024.
>
> [2] Tang X,et al. Adaptive Keyframe Sampling for Long Video Understanding. *arXiv preprint arXiv:2502.21271*, 2025.
>
> [3] Zhang S, et al. Q-Frame: Query-aware Frame Selection and Multi-Resolution Adaptation for Video-LLMs. *arXiv preprint arXiv:2506.22139*, 2025.
>
> [4] Li K, et al. MVBench: A Comprehensive Multi-modal Video Understanding Benchmark. *arXiv preprint arXiv:2311.17005*, 2023.
>
> [5] Zhang P, et al. Long Context Transfer from Language to Vision. *arXiv preprint arXiv:2406.16852*, 2024.
>
> [6] Yu S, et al. Frame-Voyager: Learning to Query Frames for Video Large Language Models. *arXiv preprint arXiv:2410.03226*, 2024.

---

> > ### Comment · Reviewer_Lg9P · 2025-11-24
> > **Thanks for the author's reply.**
> >
> > Thank you for the author's recognition and reply. However, the author's response did not completely resolve my concerns.
> >
> > Q1: As other reviewers have pointed out, the main contribution of this paper lies in its engineering aspects rather than its novelty. Although the author responded, a deeper methodological insight and experimental analysis are still lacking.
> >
> > Q2: I maintain that the statements and comparisons made regarding Table 2 in the article are still unfair. I reviewed the original Frame-Voyager paper, in which the parameter declarations and experimental results are based on LLaVa-One-Vision. However, Table 2 and the corresponding caption in the manuscript provided by the authors do not convey this information.
> >
> > Q3: As I stated earlier, `Although LLaVA-OneVision uses SigLIP-So400M-patch14-384 as the visual encoder, it fine-tunes SigLIP during the training process, which results in them having the same structure but different parameters.` The author's reply mentioned that `our current setting uses the same visual encoder for CoSELECT scoring as is used in the base MLLM for visual encoding.` It is unclear whether "same" refers to shared structure or shared parameters. Furthermore, I have doubts about whether the authors used this optimization in their current experimental setup. This requires further clarification.

---

> ### Author Response · Authors · 2025-11-24
> **Author's Response**
>
> We are grateful to the reviewer for their prompt response!
>
> > Q1: a deeper methodological insight and experimental analysis are still lacking....
>
> In addition to the **novelty contribution** discussed in our general response, we highlight that we validate our design through a comprehensive set of experiments and ablations. In this work, we study and discuss:
>
> Method design choices:
> * Varying the size of the **pre-embedding pool** of frames (Table 1, Table 9, Appendix B.4)
> * Varying the size of the **post-embedding pool** of frames (Table 10, Appendix B.5)
> * Exploring why both signals (**frame–frame similarity and text–frame similarity**) are needed (Table 5)
> * Demonstrating the effect of **each mathematical component of the algorithm** in detail (Table 7)
>   * Duration function.
>   * Composite relevance formulation.
> * Discussion of **minimal performance overhead** (Section 4.4 and Appendix C.1)
>
> Ablations:
> * Comparison with **token compression** algorithms (Table 3)
> * Varying the **architecture of the LLM backbone** (Table 4)
> * Varying the **architecture of the vision backbone** (Table 6)
> * Varying the **size of the LLM backbone** (Table 4)
> * Ablation over the **hyperparameter τ** (Table 8)
>
> We also compare across **six tasks throughout** the paper, rather than hyper-optimizing on just a few.
>
> We couldn't find any other work that explores practical design choices in such detail. We hope that all **these analyses help practitioners in making optimal technical decisions** based on their compute resources.
>
> We are happy to run any additional experiments the reviewer considers valuable in strengthening this work!
>
> > Q2: Frame-Voyager [2] clarification
>
> We apologize for not being clearer about this. We had specified the LLM size (7B) but not the backbone architecture explicitly.
> **We have now amended Table 2 to include this information.**
>
> On fairness of Table 2 more broadly: since we have limited control over architecture parity with all trained methods, we compare methods using  **same architecture where possible**, and when this is impractical, we aim for **similar backbone sizes**. This practice is consistent with reports in prior state-of-the-art work (e.g., LongVU [1], Table 1; Frame-Voyager [2], Table 1; AKS [3] - Table 1), where strict architectural parity and backbone size consistency are not enforced.
>
> > Q3: Vision encoder clarification
>
> We apologize for the lack of clarity. We use the **finetuned SigLIP-So400M-patch14-384** vision encoder—i.e., the one used in the final LLaVA-OneVision model—for all our experiments unless stated otherwise.
> **We have now clarified this in the implementation details in Section 4.1.**
>
> > References
>
> [1] Shen X, et al. LongVU: Spatiotemporal Adaptive Compression for Long Video-Language Understanding. arXiv:2410.17434, 2024.
>
> [2] Yu S, et al. Frame-Voyager: Learning to Query Frames for Video Large Language Models. arXiv:2410.03226, 2024.
>
> [3] Tang X, Qiu J, Xie L, et al. Adaptive Keyframe Sampling for Long Video Understanding. arXiv preprint arXiv:2502.21271, 2025.

---

### Official Review · Reviewer_AAbb · 2025-10-25

**Soundness:** 3
**Presentation:** 3
**Contribution:** 2
**Rating:** 6
**Confidence:** 5

**Summary:**

This paper proposes a training-free query-guided frame selection method for efficient video processing in MLLMs. It uses SigLIP cosine similarity between each frame and the given query to measure query relevance. Then it identifies scene transitions based on visual similarity. Based on these, the method adaptively allocate tokens to those scenes that is more relevant to the queries via relevance reweighting. CoSeLECT evaluates on six video understanding benchmarks and achieves state-of-the-art performance compared with both training-free and fine-tuned methods.

**Strengths:**

+ The paper writing is clear and easy to follow.
+ The method is training-free and can be applied to any LVLMs.
+ The method improves the performance on base model LLaVA-OV and Qwen2.5-VL-7B. It also outperforms other frame selection methods.

**Weaknesses:**

+ The paper should compare with more video token compression or frame selection method. For example, BOLT [1] is a frame selection method.

+ When retrained ratio goes down to 12.5%, CoSeLECT has several benchmarks lower than FastVID.

+ Although comprehensive evaluation has been down, the paper is mainly based on empirical observation and has very limited innovation.

+ The comparison does not seem entirely fair. Although 8k video tokens are finally fed into the MLLM, it still need to process additional frames during intermediate steps. Given the method involves intermediate steps and introduces computational overhead, the comparison should be made against the base model’s best performance. For example, Qwen2.5-VL got 65.1 on VideoMME.

[1] BOLT: Boost Large Vision-Language Model Without Training for Long-form Video Understanding

**Questions:**

+ When comparing with baseline models LLaVA-OV  and Qwen2.5-VL-7B, how many frames and token per frame is used within 8k context length?
+ Have you tried LLM’s text embedding instead of SigLIP text embedding?
+ Some complicated question could not be used to select key frames based on embedding similarity. For example, many questions in VideoMME are like ‘which of the statement is correct?’ Therefore, query-frame embedding similarity is not a fine-grained way for frame selection.
+ What is the computation overhead introduced and inference speed compared with base model, since the method needs to calculate similarity between consecutive frame embeddings.

---

> ### Author Response · Authors · 2025-11-21
> **Thank you for the feedback!**
>
> We thank the reviewer for their positive remarks and feedback.
>
> > On comparison with BOLT (The paper should compare.....)
>
> We thank the reviewer for their suggestion, we have added BOLT [4] to Table 1. CoSeLECT outperforms BOLT on 4/5 benchmarks (VideoMME 61.1 vs. 59.9, EgoSchema 64.8 vs. 64.0, LongVideoBench 59.3 vs. 59.6, MLVU 67.9 vs. 66.8, NextQA 80.1 vs. 79.5), achieving an average relative improvement of +1.03%. Tables 1–3 already compare CoSeLECT to more than 15 frame selection / token reduction methods. We are happy to add any additional baselines the reviewer recommends.
>
> **Changes:** We have added BOLT comparison results to Table 1.
>
> > On primarily being an empirical contribution (Although comprehensive evaluation has been down....)
>
> Thank you for bringing this up, we respond to this concern in the main response.
>
> > When retained ratio ... FastVID.
>
> FastVID operates at the token level, while CoSeLECT operates at the frame level (we drop or keep whole frames), making this comparison inherently asymmetric. Despite this handicap, CoSeLECT remains competitive: at the 12.5% setting, the two methods achieve nearly identical average performance (within 0.1 points), and CoSeLECT outperforms FastVID on 2 out of 4 benchmarks. We view this as evidence that frame-level selection can effectively match token-level compression even under aggressive constraints.
>
> > On fairness of comparison (The comparison .... Qwen2.5-VL got 65.1 on VideoMME.)
>
> Thank you for raising this point. The 65.1 VideoMME score comes from the original Qwen2.5-VL paper's evaluation of the base model. For consistency with the lmms-eval framework [5], we use Qwen2.5-VL-Instruct (the instruction-tuned variant) and re-evaluate all configurations—baseline and CoSeLECT—in a unified pipeline with matched budgets and hyperparameters.
>
> Additionally, we note that increasing the number of frames does not always improve performance. As shown in Table 10 (Appendix B.5), longer visual contexts can sometimes slightly degrade results, which motivates our evaluation within a fixed frame budget to ensure controlled comparison.
>
> > Number of frames and tokens per frame (When comparing .... context length?)
>
> For VideoMME, we use 32 frames with ~196 visual tokens per frame (≈6.3k visual tokens in total), with the remaining context reserved for text tokens.
>
> > Have you tried LLM’s text embedding instead of SigLIP text embedding?
>
> We use SigLIP for both frame and query embeddings so they lie in a shared pretrained vision–language space, enabling cosine similarity without training. LLM token embeddings are not aligned to the visual encoder; using them for frame–query similarity would either require an additional learned projection, or accessing internal hidden states from earlier layers of the MLLM for both vision and text. This would break our plug-and-play goal.
>
> > On complex questions like "which statement is correct?"(Some complicated ..... selection.)
>
> We agree that for some multiple-choice questions (e.g., “which statement is correct?” in VideoMME), query–frame similarity is less informative. In such cases, CoSeLECT effectively relies more on the visual continuity prior. This matches our ablations: the “only continuity” variant already improves over uniform sampling, and adding the text signal (full CoSeLECT) yields the best overall performance (Table 5), even when the text cue is weak.
>
> > On computation overhead and inference speed vs. the base model (What is the computation ..... frame embeddings.)
>
> We provide raw latency measurements in Appendix C.1. The dominant costs in our pipeline are similar to fps-sampling methods such as LongVU [1], AKS [2], and Q-Frame [3]: encoding frames with the vision encoder and the subsequent LLM forward pass.
>
> CoSeLECT adds only an $\mathcal{O}(N)$ similarity computation over the pre-embedded frames (pairwise vision–vision and vision–text cosine similarities). In practice, this overhead is negligible: for pre-$E_{\text{im}} = 1600$ (the worst case), the similarity computation takes **1.49 ms** out of **1.87 s** total inference time (≈**0.08%**) per query.
>
> **Changes:** We have explicitly highlighted these measurements in Appendix C.1.
>
>
> > References
>
> [1] Shen X, Xiong Y, Zhao C, et al. LongVU: Spatiotemporal Adaptive Compression for Long Video-Language Understanding. *arXiv preprint arXiv:2410.17434*, 2024.
>
> [2] Tang X, Qiu J, Xie L, et al. Adaptive Keyframe Sampling for Long Video Understanding. *arXiv preprint arXiv:2502.21271*, 2025.
>
> [3] Zhang S, Yang J, Yin J, et al. Q-Frame: Query-aware Frame Selection and Multi-Resolution Adaptation for Video-LLMs. *arXiv preprint arXiv:2506.22139*, 2025.
>
> [4] Wang Y, Xu M, Gao M, et al. BOLT: Boost Large Vision-Language Model Without Training for Long-form Video Understanding. *arXiv preprint arXiv:2503.21483*, 2025.
>
> [5] Li B, Zhang P, Zhang K, et al. LMMs-Eval: Accelerating the Development of Large Multimodal Models. *Zenodo*, v0.1.0, 2024.

---

> > ### Comment · Reviewer_AAbb · 2025-11-25
> >
> > Thanks for the author's detailed response. Most of my concerns are addressed. I have only one question in terms of the response to on fairness of comparison.
> >
> > I also try to reproduce Qwen2.5-VL's result and find it can achieve 65 on VideoMME by feeding 256 frames and control it within 20k context size by adjusting the max video pixels. So I worry about that the performance gain benefits from the selection or from adjusting the token-per-frame setting. So it is better to show with 256 frames as input after applying CoSeLECT, what will be the performance to validate the effectiveness.

---

> ### Author Response · Authors · 2025-11-27
> **Authors Follow up**
>
> We thank the reviewer for raising this point.
>
> To directly test whether the gains come from **selection** rather than changing the **token-per-frame setting**, we followed the reviewer’s recommendation and reproduced Qwen2.5-VL’s results under the suggested configuration:
>
> * Max Pixels = 90,000
> * Min Pixels = 80,000
> * Attn Implementation = flash_attention_2
> * Max Frames = 256
> * Model: **Qwen2.5-VL**
>
> Under this setup, we keep **both the number of frames (256)** and the **token-per-frame budget** fixed, and only change the **selection policy** (uniform vs. CoSeLECT). The results are:
>
> | Setting                                        |   VideoMME w/o sub   | MLVU |
> | ---------------------------------------------- | :------: | :------: |
> | Reported baseline                              |   65.1   |   71.6   |
> | Reproduced baseline (uniform frames)           |   64.8   |   71.1   |
> | + CoSeLECT (1600 → 256 frames, same max/min pixels) | **66.2** | **71.9** |
>
> **CoSeLECT improves over our reproduced baseline while keeping the frame count and token-per-frame configuration fixed**. This indicates that the gains indeed stem from **which frames are selected**, rather than from modifying the token-per-frame setting.
>
> **Changes**: We include these control experiments in the revised version (Appendix B.8)
>
> We again thank the reviewer for the constructive suggestion! Please let us know if there are any more open questions.

---

### Official Review · Reviewer_vvkc · 2025-10-27

**Soundness:** 3
**Presentation:** 2
**Contribution:** 2
**Rating:** 6
**Confidence:** 3

**Summary:**

This work introduces another method to select which frames to use for video-language understanding. This is an important topic since MLLMs often have input token limitations, and having some way to prefilter the data to find the most relevant answer can often help in increasing the results. The method is training-free and can be plugged with different models. Their main competitor is AKS (Adaptive Keyframe Sampling) that also adopts a training-free paradigm. However, there are some differences with the method proposed here. Whereas AKS splits the video into equal halves, CoSeLECT segments the video into different lengths depending on intra-clip similarity. The method introduced by the authors seems a bit more flexible than AKS while providing slightly better results. The authors evaluate their method across several common benchmarks such as NextQA, MLVU, VideoMME, MVbench, and LongVideoBench. They show that their method is competitive with training-based methods such as LongVu. They also compare their method with different token reduction techniques such as uniform sampling, VisionZip, PruneVid, and others, for which they also have competitive results.

**Strengths:**

- A simple and yet effective training-free method for frame selection
- Adaptative and not relying on specific video sub-clip size
- Paper well-written
- Extensive comparison with similar methods such as AKS
- Extension comparison with both training-based and training-free frame selection method
- Good ablation over the different hyper-parameters such as similarity threshold or frame pool size

**Weaknesses:**

- Some overhead introduced by the method since frames need to be processed through a SigLip encoder to compute frame similarity. Depending on the number of frames being processed, this can have an important impact even if this operation can be parallelised.
- Lack of ablation over the vision and text encoder.
- The paper title and abstract does not exactly match the ones in OpenReview (don't know how much that can be an issue or not)

**Questions:**

Why choosing SigLIP-ViT and not another model? Did you perform an ablation on those?

---

> ### Author Response · Authors · 2025-11-21
> **Thank you for the feedback!**
>
> We thank the reviewer for their positive remarks! We hope the following addresses any open questions.
>
> > Performance overhead
>
> We provide raw latency measurements in Appendix C.1. The inference time for the baseline(No selection mechanism) is $0.32$ ms per query on average.
>
> The dominant costs in our pipeline are similar to fps-sampling methods such as LongVU [1], AKS [2], and Q-Frame [3]: encoding frames with the vision encoder and the subsequent LLM forward pass.
>
> CoSeLECT itself adds only an $\mathcal{O}(N)$ similarity computation over the pre-embedded frames (pairwise vision–vision and vision–text cosine similarities). In practice, this overhead is negligible: for pre-$E_{\text{im}} = 1600$ (the worst case), the similarity computation takes **1.49 ms** out of **1.87 s** total inference time (≈**0.08%**) per query.
>
> **Changes Applied:** We have explicitly highlighted these measurements in Appendix C.1 .
>
> > Lack of ablation over the vision and text encoder
>
> Table 4 explores different **vision encoders**, and Table 6 explores different **language encoders**. We observe consistent improvements across proprietary models (GPT-4o) and open-source language encoders of different sizes (0.5B and 7B) and training regimes (LLaVA-OV vs. Qwen2.5-VL).
>
> On the vision side(Table 4), we evaluate three encoders with varying sizes(151M and 878M) and find that CoSeLECT improves performance in all cases. The absolute scores vary slightly with encoder size, which we view as a natural accuracy–compute trade-off that can be chosen by the end user.
>
> We are happy to add any further comparisons that the reviewer sees fit!
>
> > References
>
> [1] Shen X, Xiong Y, Zhao C, et al. LongVU: Spatiotemporal Adaptive Compression for Long Video-Language Understanding. *arXiv preprint arXiv:2410.17434*, 2024.
>
> [2] Tang X, Qiu J, Xie L, et al. Adaptive Keyframe Sampling for Long Video Understanding. *arXiv preprint arXiv:2502.21271*, 2025.
>
> [3] Zhang S, Yang J, Yin J, et al. Q-Frame: Query-aware Frame Selection and Multi-Resolution Adaptation for Video-LLMs. *arXiv preprint arXiv:2506.22139*, 2025.

---

### Official Review · Reviewer_CaVy · 2025-10-31

**Soundness:** 3
**Presentation:** 3
**Contribution:** 3
**Rating:** 4
**Confidence:** 5

**Summary:**

This paper proposes a training-free method that can be seamlessly integrated into existing multimodal large language models (MLLMs). The approach jointly considers both frame-level visual diversity and overall video length, leading to more balanced and informative video representations. The method achieves state-of-the-art performance across multiple video understanding benchmarks, demonstrating both simplicity and effectiveness.

**Strengths:**

1. The proposed method is simple but effective, requiring no additional training while significantly improving performance.

2. The design is model-agnostic and can be easily plugged into various MLLM architectures, indicating strong generality and practical utility.

3. Experimental results are convincing and comprehensive, covering multiple datasets and metrics, with clear visualizations that illustrate the method’s contribution.

4. The paper is well-written and easy to follow, making the technical insights accessible.

**Weaknesses:**

1. The main concern lies in the limited novelty of the method. The use of text–visual embedding similarity as a selection strategy is not conceptually new and has been widely seen in prior works as an auxiliary component or ablation. While the empirical results are strong, the contribution is mainly engineering-level, lacking deeper methodological insight or theoretical advancement.

2. In addition, the paper does not clearly explain how the method mitigates temporal information loss when modeling long videos.

**Questions:**

Please see the weaknesses.

---

> ### Author Response · Authors · 2025-11-21
> **Thank you for the feedback!**
>
> We thank the reviewer for their positive feedback, and we hope the following is helpful. Please let us know if we can do any additional clarifications.
>
> > “The main concern lies in the limited novelty of the method. The use of text–visual embedding similarity as a selection strategy is not conceptually new and has been widely seen in prior works as an auxiliary component or ablation. While the empirical results are strong, the contribution is mainly engineering-level, lacking deeper methodological insight or theoretical advancement.”
>
> To avoid repetition of text, we address this best as we can in the "Theoretical Novelty" section of our main response.
>
>
> >“In addition, the paper does not clearly explain how the method mitigates temporal information loss when modeling long videos.”
>
> We appreciate this concern and clarify that CoSeLECT is explicitly designed to mitigate temporal information loss under a fixed context budget:
>
> 1. We first form **temporally coherent subclips** using a visual continuity signal, so selection is performed over segments that already respect local temporal structure instead of treating frames as independent.
> 2. We then **allocate the global frame budget across segments** using both relevance and duration, ensuring that long and relevant segments receive more frames and are not underrepresented.
> 3. Within each segment, we **spread selected frames over time** by dividing the segment into slices and choosing the most text-relevant frame per slice, preventing all selected frames from collapsing around a single peak moment.
>
> Ultimately, the hard limit for very long videos is imposed by the LLM’s context length. CoSeLECT’s goal is to use this limited budget as **temporally faithfully and query-aware as possible**. The improvements we observe on long-horizon benchmarks suggest that this segmentation, budgeting, and intra-segment spreading strategy preserves temporal structure more effectively than existing training-free baselines.

---

### Author Response · Authors · 2025-11-21
**General Remarks by Authors**

We thank all reviewers for their thoughtful and positive feedback. We are particularly encouraged that they found CoSeLECT:

* **Simple, training-free, and broadly applicable.** It is described as *“simple but effective”* and *“training-free”*, *“can be seamlessly integrated into existing MLLMs”*, *“model-agnostic”*, *“can be applied to any LVLMs”*, and a *“lightweight, plug-and-play solution… useful for real-world deployment”* (CaVy, vvkc, AAbb, Lg9P).

* **Empirically strong and well evaluated.** Reviewers note that CoSeLECT *“achieves state-of-the-art performance across multiple video understanding benchmarks”*, *“better performance than existing training-free methods”*, is *“more flexible than AKS… competitive with training-based methods such as LongVu”*, and *“state-of-the-art… compared with both training-free and fine-tuned methods”*, with *“convincing and comprehensive”* experiments, *“extensive comparison”*, *“good ablation”*, and *“evaluation on six video understanding benchmarks”* (CaVy, vvkc, AAbb, Lg9P).

* **Clear and reproducible.** The paper is called *“well-written and easy to follow”* and *“well-described, easy to reproduce”*, with *“clear visualizations that illustrate the method’s contribution”* (CaVy, vvkc, AAbb, Lg9P).

The requested changes mainly led to focused clarifications and modest additions, rather than major overhauls, which we take as a positive signal that the core approach and empirical findings were already in solid shape.

Below, we address the main concerns regarding novelty and computational overhead in detail.

* **Theoretical Novelty.**
  We agree that using query–frame and frame–frame similarity as *signals* is not new. Our contribution is not a new similarity measure, but a **concrete, training-free allocation scheme** that specifies *how* to fuse these signals into a single-stage, query-aware budgeting mechanism over temporally coherent segments. CoSeLECT:
  1. **Segments by visual continuity**, so selection operates over coherent temporal units instead of isolated frames;
  2. Computes a **composite segment score** (max + mean text–frame similarity);
  3. Uses a **duration-aware frame budget** and then **text-guided selection within segments**.

  Ablations (Table 5, Appendix B) show that removing any component (only text, only continuity, no duration term, or simpler scoring) consistently harms performance, indicating that the contribution lies in the **overall allocation pipeline**, not a single heuristic.

  Beyond this design, CoSeLECT is **empirically strong**, leading in 5/6 benchmarks among training-free methods and outperforming fully trained approaches such as LongVU [1] under the same backbone. It is **decoupled from any specific vision encoder or LLM**, showing consistent gains across backbones, and has a **straightforward implementation** with ablations that make its behavior and trade-offs transparent. Finally, we expose and systematically study **interpretable knobs**—the size of the pre-embedding frame pool, the number of frames fed to the LLM, and the choice of LLM / vision backbones—providing actionable guidance for practitioners (e.g., Section B.4 shows that *“more FPS is not always better”* once selection is well designed).

* **Computational overhead.**
  While CoSeLECT does require encoding a large pool of frames, this is **in line with existing state-of-the-art methods** such as AKS [2], LongVU [1], BOLT [4], and other FPS-based approaches: any method that must *reason* about which frames to keep generally needs to encode a sizeable candidate pool. In practice, the **dominant cost** is shared with these methods: running the vision encoder and the subsequent LLM forward pass.

  The *additional* cost introduced by CoSeLECT is only an $O(N)$ similarity computation over the pre-embedded frames (pairwise vision–vision and vision–text cosine similarities). As quantified in Appendix C.1, for the largest setting with pre-$E_{\text{im}} = 1600$ frames, the similarity computation takes **1.49 ms** out of **1.87 s** total inference time (≈**0.08%**) per query. Thus, while CoSeLECT operates over a larger candidate pool, the **extra overhead due to the selection logic itself is negligible** compared to the shared cost of vision and LLM inference.

---

#### References

[1] Shen X, Xiong Y, Zhao C, et al. LongVU: Spatiotemporal Adaptive Compression for Long Video-Language Understanding. *arXiv preprint arXiv:2410.17434*, 2024.
[2] Tang X, Qiu J, Xie L, et al. Adaptive Keyframe Sampling for Long Video Understanding. *arXiv preprint arXiv:2502.21271*, 2025.
[3] Zhang S, Yang J, Yin J, et al. Q-Frame: Query-aware Frame Selection and Multi-Resolution Adaptation for Video-LLMs. *arXiv preprint arXiv:2506.22139*, 2025.
[4] Wang Y, Xu M, Gao M, et al. BOLT: Boost Large Vision-Language Model Without Training for Long-form Video Understanding. *arXiv preprint arXiv:2503.21483*, 2025.

---

### Author Response · Authors · 2025-12-04
**For the AC: A Summary of the Rebuttal Session**

### **Paper Summary**

CoSeLECT offers a lightweight, training-free, and broadly applicable method for **adaptive frame selection** that:

* Is **empirically strong across six benchmarks**, often outperforming existing training-free methods and being competitive with or better than trained approaches.
* **Integrates seamlessly into existing video-LLMs**, without retraining, and is decoupled from specific LLM or vision backbones.
* Is backed by a **thorough analysis** of design choices and trade-offs, providing **practical, actionable guidance** for deploying long-video MLLMs under realistic compute budgets.

### **Reviewer Outcomes**

**The responses were mainly very positive; reviewer questions focused on clarifications or small ablations rather than major changes/experiments.**

* Reviewer **AAbb** (Score:**6**, Responded once during rebuttal) explicitly stated that *their concerns were mostly resolved* after the first author response, with only one small followup clarification, which is fully answered.
* Reviewer **Lg9P** (Score:**4**, Responded once during rebuttal) engaged in an extensive review; we provided detailed responses to all questions and clarifications raised. No unresolved questions remain.
* Reviewers **vvkc** (Score:**6**, No responses in rebuttal period) and **CaVy** (Score:**4**, No responses in rebuttal period) each had a single technical clarification respectively, which we addressed fully in the rebuttal; no new concerns were raised.

*All reviewer recommended changes have been added to the most recent draft of the paper.*

### **Positive Comments**

Across all reviewers, the core assessment of CoSeLECT was consistently positive on three fronts:

* **Simple, training-free, and broadly applicable.**
   CoSeLECT is repeatedly described as “simple but effective”, “training-free”, “model-agnostic”, “lightweight, plug-and-play”, and “seamlessly integrable into existing MLLMs / LVLMs” (CaVy, vvkc, AAbb, Lg9P), with reviewers emphasizing its practicality for real-world deployment.

* **Empirically strong and well evaluated.**
   Reviewers highlight that CoSeLECT “achieves state-of-the-art performance across multiple video understanding benchmarks”, “outperforms existing training-free methods”, is “more flexible than AKS and competitive with training-based methods such as LongVU”, and is “state-of-the-art compared with both training-free and fine-tuned methods”. They praise the experiments as “convincing and comprehensive”, with “extensive comparisons”, “good ablations”, and evaluation on six benchmarks (CaVy, vvkc, AAbb, Lg9P).

* **Clear, reproducible presentation.**
   The paper is described as “well-written and easy to follow” and “well-described, easy to reproduce”, with “clear visualizations that illustrate the method’s contribution” (CaVy, vvkc, AAbb, Lg9P).

### **Main Concerns**

**1\. Perceived lack of theoretical novelty**

Reviewers commented that using text and visual similarity as signals is not conceptually new.

In our response, we clarified that the contribution is not a new similarity function, but a training-free, query-aware ***frame budget allocation scheme*** that leverages similarity as a part of the algorithm. CoSeLECT combines continuity-based segmentation, segment-level relevance scoring, and duration-aware budgeting into a **single, practical pipeline** that can be dropped into existing video-LLMs without retraining.

Our contribution is strengthened by **systematic empirical evidence**: 10+ ablations and in-depth comparison with related work. Ablations demonstrate the **importance of every element of CoSeLECT’s algorithm**, and that it is **robust across backbones, model sizes, and token-compression** baselines.

To our knowledge, **no prior work explores this design space for training-free frame selection with such rigor**, and we position these analyses as **practical guidance** for deploying frame selection on video MLLMs under realistic compute budgets.

**2\. Computational overhead**

Some reviewers expressed hesitation about the cost of encoding a large pool of frames.

We clarified that this overhead is shared with existing state-of-the-art methods (AKS \[2\], LongVU \[1\], BOLT \[3\], and other FPS-based approaches): any method that reasons about which frames to keep must process a sizable candidate pool.

The *additional* cost specific to CoSeLECT **is only an (O(N)) similarity computation over pre-embedded frames**. Practically this accounts for a negligible ≈**0.08%** increase in inference time per query in the worst case (1600 frames). *We have added the detailed computational analysis to the appendix (C.1). We thank reviewers for asking us about this, as these new numbers bolster our efficiency claims.*

**Overall, reviewers are largely aligned on the strength of the results, the practicality and elegance of the approach, and the breadth of ablations.**

---

> ### Author Response · Authors · 2025-12-04
> **References**
>
> [1] Shen X, Xiong Y, Zhao C, et al. LongVU: Spatiotemporal Adaptive Compression for Long Video-Language Understanding. arXiv preprint arXiv:2410.17434, 2024.
>
> [2] Tang X, Qiu J, Xie L, et al. Adaptive Keyframe Sampling for Long Video Understanding. arXiv preprint arXiv:2502.21271, 2025.
>
> [3] Wang Y, Xu M, Gao M, et al. BOLT: Boost Large Vision-Language Model Without Training for Long-form Video Understanding. arXiv preprint arXiv:2503.21483, 2025.

---

### Meta-Review · Area_Chair_cmvb · 2026-01-12

**Summary:**

Overall, reviewers found the proposed CoSeLECT framework to be simple, training-free, and empirically competitive across multiple benchmarks. However, multiple reviewers raised concerns that the contribution is primarily engineering-driven with limited methodological novelty, since the main signals (text–frame relevance and frame–frame similarity) are not new. In addition, reviewers questioned the clarity and fairness of certain experimental comparisons (especially against end-to-end trained systems), and requested stronger justification of the “lightweight” claim given the need to pre-encode large frame pools. While the rebuttal provided additional clarifications and overhead analysis, I believe the paper is not yet ready for publication at ICLR, and therefore I recommend rejection.

**Reviewer Concerns:**

Reviewer CaVy (Rating: 4)

Addressed: Clarified how CoSeLECT mitigates temporal information loss (segmentation + duration-aware budgeting + intra-segment spreading).

Outstanding: Concern that the work has limited novelty and is largely engineering-level, lacking deeper methodological insight.

Reviewer vvkc (Rating: 6)

Addressed: Overhead/latency concern addressed with added runtime analysis; clarified encoder ablations (vision/text) and rationale for SigLIP.

Outstanding: Minor: presentation/clarity issues (e.g., title/abstract mismatch) are not central; overall concerns largely resolved.

Reviewer AAbb (Rating: 6)

Addressed: Added stronger baseline comparisons (incl. BOLT), clarified fairness, and provided control experiment showing gains persist even with fixed 256 frames + fixed token-per-frame (selection effect).

Outstanding: Remaining concern is primarily limited innovation (method seen as empirical/engineering-heavy), though reviewer largely satisfied.

Reviewer Lg9P (Rating: 4)

Addressed: Added latency analysis; clarified Table 2 setup (Frame-Voyager backbone), fixed notation/clarity issues, added limitations section and extra comparisons.

Outstanding: Reviewer remains unconvinced on novelty, and still raises concern about fairness/clarity of Table 2 comparisons and ambiguity around “shared encoder reuse” claims (shared parameters vs. structure).

**Reviewer Scores:**

see above.

---

### Decision · Program_Chairs · 2026-01-26

Reject